# Regulation of inorganic carbon acquisition in a red tide alga (*Skeletonema costatum*): the importance of phosphorus availability

Guang Gao[a,b,c],Jianrong Xia[a*], Jinlan Yu[a], Jiale Fan[b], Xiaopeng Zeng[a]

[a]School of Environmental Science and Engineering, Guangzhou University, Guangzhou, 510006, China

[b]Jiangsu Key Laboratory of Marine Bioresources and Environment, Huaihai Institute of Technology, Lianyungang, 222005, China

[c]State Key Laboratory of Marine Environmental Science, Xiamen University, Xiamen 361005, China

[*]Corresponding author, Email: jrxia@gzhu.edu.cn; Phone: +86 (0)20 39366941; Fax: +86 (0)20 39366949

**Abstract:**

*Skeletonema costatum* is a common bloom-forming diatom and encounters eutrophication and severe carbon dioxide ($CO_2$) limitation during red tides. However, little is known regarding the role of phosphorus (P) in modulating inorganic carbon acquisition in *S. costatum*,particularly under $CO_2$ limitation conditions. We cultured *S. costatum* under five phosphate levels (0.05, 0.25, 1, 4, 10 $\mu$mol $L^{-1}$) and then treated it with two $CO_2$ conditions (2.8 and 12.6 $\mu$mol $L^{-1}$) for two hours. The lower $CO_2$ reduced net photosynthetic rate at lower phosphate levels (< 4 $\mu$mol $L^{-1}$) but did not affect it at higher phosphate levels (4 and 10 $\mu$mol $L^{-1}$). In contrast, the lower $CO_2$ induced higher dark respiration rate at lower phosphate levels (0.05 and 0.25 $\mu$mol $L^{-1}$) and did not affect it at higher phosphate levels (> 1 $\mu$mol $L^{-1}$). The lower $CO_2$ did not change relative electron transport rate (rETR) at lower phosphate levels (0.05 and 0.25 $\mu$mol $L^{-1}$) and increased it at higher phosphate levels (> 1 $\mu$mol $L^{-1}$). Photosynthetic $CO_2$ affinity ($1/K_{0.5}$) increased with phosphate levels. The lower $CO_2$ did not affect photosynthetic $CO_2$ affinity at 0.05 $\mu$mol $L^{-1}$ phosphate but enhanced it at the other phosphate levels. Activity of extracellular carbonic anhydrase was dramatically induced by the lower $CO_2$ at phosphate replete conditions (> 0.25 $\mu$mol $L^{-1}$) and the same pattern also occurred for redox activity of plasma membrane. Direct bicarbonate ($HCO_3^-$) use was induced when phosphate concentration was more than 1 $\mu$mol $L^{-1}$. These findings indicate P enrichment could enhance inorganic carbon acquisition and thus maintain the photosynthesis rate in *S. costatum* grown under $CO_2$ limiting conditions via increasing activity of extracellular carbonic anhydrase and facilitating direct $HCO_3^-$ use. This study sheds light on how bloom-forming algae cope with carbon limitation during the development of red tides.

23    **Keywords:** carbonic anhydrase; $CO_2$ concentrating mechanisms; pH compensation point;

24    photosynthesis; redox activity; respiration

## 1. Introduction

Diatoms are unicellular photosynthetic microalgae that can be found worldwide in

freshwater and oceans. Marine diatoms account for 75% of the primary productivity for

coastal and other nutrient-rich zones and approximately 20% of global primary production

(Field *et al.*, 1998; Falkowski, 2012), hence playing a vital role in the marine biological

carbon pump as well as the biogeochemical cycling of important nutrients, such as nitrogen

and silicon (Nelson *et al.*, 1995; Moore *et al.*, 2013; Young & Morel, 2015). Diatoms usually

dominate the phytoplankton communities and form large-scale blooms in nutrient–rich zones

and upwelling regions (Bruland *et al.*, 2001; Anderson *et al.*, 2008; Barton *et al.*, 2016).

Nutrient enrichment is considered as a key factor that triggers algal blooms albeit the

occurrence of diatom blooms may be modulated by other environmental factors, such as

temperature, light intensity, salinity, and so forth (Smetacek & Zingone, 2013; Jeong *et al.*,

2015). When inorganic nitrogen and phosphorus are replete, diatoms can out-compete

chrysophytes, raphidophytes and dinoflagellates (Berg *et al.*, 1997; Jeong *et al.*, 2015; Barton

*et al.*, 2016) and dominate algal blooms due to their quicker nutrient uptake and growth rate.

In normal natural seawater (pH 8.1, salinity 35), $HCO_3^-$ is the majority (~90%) of total

dissolved inorganic carbon (DIC, 2.0–2.2 mM). $CO_2$ (1%, 10–15 μM), which is the only

direct carbon source that can be assimilated by all photosynthetic organisms, only accounts

for 1% of total dissolved inorganic carbon. Diatoms' ribulose-1,5-bisphosphate

carboxylase/oxygenase (Rubisco), catalyzing the primary chemical reaction by which $CO_2$ is

transformed into organic carbon, has a relatively low affinity for $CO_2$ and is commonly less

than half saturated under current $CO_2$ levels in seawater (Hopkinson & Morel, 2011),

suggesting that $CO_2$ is limiting for marine diatoms' carbon fixation. To cope with the $CO_2$
limitation in seawater and maintain a high carbon fixation rate under the low $CO_2$ conditions,
diatoms have evolved various inorganic carbon acquisition pathways and $CO_2$ concentrating
mechanisms (CCMs), for instance, active transport of $HCO_3^-$, the passive influx of $CO_2$,
multiple carbonic anhydrase (including both common (α, β, γ, found in all algae) and unusual
(δ, ζ, found only in diatoms) families that carries out the fast interconversion of $CO_2$ and
$HCO_3^-$), assumed C4-type pathway (using phosphoenolpyruvate to capture more $CO_2$ in the
periplastidal compartment), to increase the concentration at the location of Rubisco and thus
the carbon fixation. (Hopkinson & Morel, 2011; Hopkinson *et al.*, 2016). *Skeletonema*
*costatum* is a worldwide diatom species that can be found from equatorial to polar waters. It
usually dominates large-scale algal blooms in eutrophic seawaters (Wang, 2002; Li *et al.*,
2011). When blooms occur, seawater pH increases and $CO_2$ decreases because the dissolution
rate of $CO_2$ from the atmosphere cannot catch up with its removal rate caused by intensive
photosynthesis of algae. For instance, pH level in the surface waters of the eutrophic Mariager
Fjord, Denmark, could be up to 9.75 during algal blooms (Hansen, 2002). Consequently, *S.*
*costatum* experiences very severe $CO_2$ limitation when blooms occur. To deal with it, *S.*
*costatum* has developed multiple CCMs (Nimer *et al.*, 1998; Rost *et al.*, 2003). However,
contrasting findings were reported. Nimer *et al.* (1998) documented that extracellular carbonic
anhydrase activity in *S. costatum* was only induced when $CO_2$ concentration was less than 5
μmol $L^{-1}$ while Rost *et al.* (2003) reported that activity of extracellular carbonic anhydrase
could be detected even when $CO_2$ concentration was 27 μmol $L^{-1}$. Chen and Gao (2004)
showed that in *S. costatum* had little capacity in direct $HCO_3^-$ utilization. On the other hand,
Rost *et al.* (2003) demonstrated that this species could take up $CO_2$ and $HCO_3^-$
simultaneously.

71        Phosphorus (P) is an indispensable element for all living organisms, serving as an integral

component of lipids, nucleic acids, adenosine-triphosphate (ATP) and a diverse range of other
metabolites. Levels of bioavailable phosphorus are very low in many ocean environments and
phosphorus enrichment can commonly increase algal growth and marine primary productivity
in the worldwide oceans (Davies & Sleep, 1989; Müller & Mitrovic, 2015; Lin *et al.*, 2016).
Due to the essential role of phosphorus, extensive studies have been conducted to investigate
the effect of phosphorus on photosynthetic performances (Geider *et al.*, 1998; Liu *et al.*, 2012;
Beamud *et al.*, 2016), growth (Jiang *et al.*, 2016; Reed *et al.*, 2016; Mccall *et al.*, 2017),
phosphorus acquisition, utilization and storage (Lin *et al.*, 2016; Gao et al., 2018a). Some
studies show the essential role of phosphorus in regulating inorganic carbon acquisition in
green algae (Beardall *et al.*, 2005; Hu & Zhou, 2010). In terms of *S. costatum*, studies
regarding the inorganic carbon acquisition in *S. costatum* focus on its response to variation of
$CO_2$ availability. The role of phosphorus in *S. costatum'*s CCMs remains unknown. Based on
the connection between phosphorus and carbon metabolism in diatoms (Brembu *et al.*, 2017),
we hypothesize that phosphorus enrichment could enhance inorganic carbon utilization and
hence maintain high rates of photosynthesis and growth in *S. costatum* under $CO_2$ limitation
conditions. In the present study, we aimed to test this hypothesis by investigating the variation
of CCMs (including active transport of $HCO_3^-$ and carbonic anhydrase activity) and
photosynthetic rate under five levels of phosphate and two levels of $CO_2$ conditions. We also
measured redox activity of plasma membrane as it is deemed to be critical to activate carbonic

anhydrase (Nimer *et al.*, 1998). Our study ~~would~~ provides helpful insights into how

bloom-forming diatoms overcome $CO_2$ limitation to maintain a quick growth rate during red

tides.

**2.  Materials and Methods**

*2.1. Culture conditions*

   *Skeletonema costatum* (Grev.) Cleve from Jinan University, China, was cultured in f/2

artificial seawater with five phosphate levels (0.05, 0.25, 1, 4, 10 μmol $L^{-1}$) by adding

different amounts of $NaH_2PO_4$ $2H_2O$. The cultures were carried out semi-continuously at

$20^{o}C$ for seven days. The light irradiance was set as 200 μmol photons $m^{-2} s^{-1}$, with a light

and dark period of 12: 12. The cultures were aerated with ambient air (0.3 L $min^{-1}$) to

maintain the pH around 8.2. The cells during exponential phase were collected and rinsed

twice with DIC-free seawater that was made according to Xu *et al.* (2017). Afterwards, cells

were resuspended in fresh media with two levels of pH (8.20 and 8.70, respectively

corresponding to ambient $CO_2$ (12.6 μmol $L^{-1}$~~, AC~~) and low $CO_2$ (2.8 μmol $L^{-1}$~~, LC~~) under

corresponding phosphate levels for two hours before the following measurements, with a cell

density of $1.0 \times 10^6$ $mL^{-1}$. The concentrations of DIC were 2109 ±36 and 1802 ±38 μmol (kg

seawater)$^{-1}$, respectively. Cell density was determined by direct counting with an improved

Neubauer haemocytometer (XB-K-25, Qiu Jing, Shanghai, China). This transfer aimed to

investigate the effects of phosphate on DIC acquisition under a $CO_2$ limitation condition. The

pH of 8.70 was chosen considering that it is commonly used as a $CO_2$ limitation condition

(Nimer *et al.*, 1998; Chen & Gao, 2004) and also occurs during algal blooms (Hansen, 2002).

Two hours should be enough to activate CCMs in *S. costatum* (Nimer *et al.*, 1998). The cell

density did not vary during the two hours of pH treatment. All experiments were conducted in
triplicates.
*2.2. Manipulation of seawater carbonate system*
The two levels of pH (8.20 and 8.70) were obtained by aerating the ambient air and pure
nitrogen (99.999%) ~~till~~until the target value, and were then maintained with a buffer of 50
mM tris (hydroxymethyl) aminomethane-HCl. The cultures were open to the ambient
atmosphere and the rise of culture pH was due to algal photosynthesis below 0.02 unit~~e~~s
(corresponding to the ~~rise~~ decrease of $CO_2$ less than 0.7 and 0.2 μmol $L^{-1}$ for pH 8.20 and 8.70
treatments, respectively) during the two hours of pH treatment. $CO_2$ level in seawater was
calculated via CO2SYS (Pierrot *et al.*, 2006) based on measured pH and TAlk, using the
equilibrium constants of K1 and K2 for carbonic acid dissociation (Roy *et al.*, 1993) and the
$KSO_4^-$ dissociation constant from Dickson (1990).
~~2.3.~~ The $pH_{NBS}$ was measured by a pH meter (pH 700, Eutech Instruments, Singapore) that
was equipped with an Orion[®] 8102BN Ross combination electrode (Thermo Electron Co.,
USA) and calibrated with standard National Bureau of Standards (NBS) buffers (pH = 4.01,
7.00, and 10.01 at 25.0 °C; Thermo Fisher Scientific Inc., USA). Total alkalinity (TAlk) was
determined at 25.0 °C by Gran acidimetric titration on a 25-ml sample with a TAlk analyzer
(AS-ALK1, Apollo SciTech, USA), using the precision pH meter and an Orion[®] 8102BN
Ross electrode for detection. To ensure the accuracy of TAlk, the TAlk analyser was regularly
calibrated with certified reference materials from Andrew G. Dickson's laboratory (Scripps
Institute of Oceanography, U.S.A.) at a precision of ±2 μmol $kg^{-1}$. ~~$CO_2$ level in seawater was~~
~~calculated via CO2SYS (Pierrot *et al.*, 2006) based on measured pH and TAlk, using the~~
~~equilibrium constants of K1 and K2 for carbonic acid dissociation (Roy *et al.*, 1993) and the~~
~~KSO$_4^-$ dissociation constant from Dickson (1990).~~

*2.3. Chlorophyll fluorescence measurement*

Chlorophyll fluorescence was measured with a pulse modulation fluorometer (PAM-2100,
Walz, Germany) to assess electron transport in photosystem II (the first protein complex in the
light-dependent reactions of photosynthesis) and the possible connection between electron
transport and redox activity of the plasma membrane. The measuring light and actinic light
were 0.01 and 200 μmol photons m$^{-2}$ s$^{-1}$, respectively. The saturating pulse was set 4,000 μmol
photons m$^{-2}$ s$^{-1}$ (0.8 s). ~~Relative e~~Electron transport in photosystem II (~~r~~ETR, μmol e$^-$ (mg Chl
*a*)$^{-1}$~~$^2$~~ s$^{-1}$) ~~=~~= 0.5 ×E ×$\Phi_{PSII}$ ×$\bar{a}^*$ (Dimier *et al.*, 2009; Alderkamp *et al.*, 2012), where E (μmol
photons m$^{-2}$ s$^{-1}$) is the ambient light density, $\Phi_{PSII}$ (dimensionless) is the PSII photochemical
efficiency and $\bar{a}^*$ is Chl *a*–specific absorption coefficient (m$^{-2}$ (mg Chl *a*)$^{-1}$). Since $\bar{a}^*$ is
light-dependent, we used the value of 0.0138 based on Lefebvre *et al*'s (2007) study in which
the light density is very close to ours.
~~(F$_M$' – F$_t$) / F$_M$' × 0.5 × PFD (Gao et al., 2018), where F$_M$' is the maximal fluorescence~~
~~levels from algae in the actinic light after application a saturating pulse, Ft is the fluorescence~~
~~at an excitation level and PFD is the actinic light density.~~

*2.4. Estimation of photosynthetic oxygen evolution and respiration*

The net photosynthetic and respiration rates of *S. costatum* were measured using a
Clark-type oxygen electrode (YSI Model 5300, USA) that was held in a circulating water bath
(Cooling Circulator; Cole Parmer, Chicago, IL, USA) to keep the setting temperature (20$^o$C).
Five mL of samples were transferred to the oxygen electrode cuvette and were stirred during
measurement. The light intensity and temperature were maintained as the same as that in the
growth condition. The illumination was provided by a halogen lamp. The increase of oxygen
content in seawater within five minutes was defined as net photosynthetic rate. To measure
dark respiration rate, the samples were placed in darkness and the decrease of oxygen content
within ten minutes was defined as dark respiration rate given the slower oxygen variation rate
for dark respiration. Net photosynthetic rate and dark respiration rate were presented as μmol
$O_2 (10^9 \text{cells})^{-1} \text{h}^{-1}$.
To obtain the curve of net photosynthetic rate versus DIC, seven levels of DIC (0, 0.1, 0.2,
0.5, 1, 2, and 4 mM) were made by adding different amounts of $NaHCO_3$ to the Tris buffered
DIC-free seawater (pH 8.20). The algal samples were washed twice with DIC-free seawater
before transferring to the various DIC solutions. Photosynthetic rates at different DIC levels
were measured under saturating irradiance of 400 μmol photons $\text{m}^{-2} \text{s}^{-1}$ and growth
temperature. The algal samples were allowed to equilibrate for 2–3 min at each DIC level
during which period a linear change in oxygen concentration was obtained and recorded. The
parameter, photosynthetic half saturation constant ($K_{0.5}$, i.e., the DIC concentration required to
give half of DIC-saturated maximum rate of photosynthetic $O_2$ evolution), was calculated
from the Michaelis-Menten kinetics equation (Caemmerer and Farquhar 1981): $V = V_{max} \times [S]$
$/ (K_{0.5} + [S])$, where V is the real-time photosynthetic rate, $V_{max}$ is maximum photosynthetic
rate and [S] is the DIC concentration. The value of $1/ K_{0.5}$ represents photosynthetic DIC
affinity. $K_{0.5}$ for $CO_2$ was calculated via CO2SYS (Pierrot *et al.*, 2006) based on pH and TA,
using the equilibrium constants of K1 and K2 for carbonic acid dissociation (Roy *et al.*, 1993)
and the $KSO_4^-$ dissociation constant from Dickson (1990).
*2.5. Measurement of photosynthetic pigment*
To determine the photosynthetic pigment (Chl *a*) content, 50 mL of culture were filtered
on a Whatman GF∕F filter, extracted in 5 mL of 90% acetone for 12 h at $4^o$C, and centrifuged
(3, 000 *g*, 5 min). The optical density of the supernatant was scanned from 200 to 700 nm
with a UV-VIS spectrophotometer (Shimadzu UV-1800, Kyoto, Japan). The concentration of
Chl *a* was calculated based on the optical density at 630 and 664 nm: Chl *a* $=11.47 \times OD_{664} -$
$0.40 \times OD_{630}$ (Gao et al., 2018b), and was normalized to pg cells$^{-1}$.
*2.6.  Measurement of extracellular carbonic anhydrase activity*
Carbonic anhydrase activity was assessed using the electrometric method (Gao *et al.*,
2009). Cells were harvested by centrifugation at 4, 000 *g* for five minutes at $20^o$C, washed
once and resuspended in 8 mL Na-barbital buffer (20 mM, pH 8.2). Five mL $CO_2$-saturated
icy distilled water was injected into the cell suspension, and the time required for a pH
decrease from 8.2 to 7.2 at $4^o$C was recorded. Extracellular carbonic anhydrase ($CA_{ext}$)
activity was measured using intact cells. CA activity (E.U.) was calculated using the
following formula: E.U. $= 10 \times (T_0 / T - 1)$, where $T_0$ and T represent the time required for the
pH change in the absence or presence of the cells, respectively.
*2.7.  Measurement of redox activity in the plasma membrane*
The redox activity of plasma membrane was assayed by monitoring the change in
$K_3Fe(CN)_6$ concentration that accompanied reduction of the ferricyanide to ferrocyanide. The
ferricyanide [$K_3Fe(CN)_6$] cannot penetrate intact cells and has been used as an external
electron acceptor (Nimer *et al.*, 1998; Gao *et al.*, 2018b). Stock solutions of $K_3Fe(CN)_6$ were
freshly prepared before use. Five mL of samples were taken after two hours of incubation
with 500 μmol $K_3Fe(CN)_6$ and centrifuged at 4000 g for 10 min ($20^o$C). The concentration of
$K_3Fe(CN)_6$ in the supernatant was measured spectrophotometrically at 420 nm (Shimadzu
UV-1800, Kyoto, Japan). The decrease of $K_3Fe(CN)_6$ during the two hours of incubation
was used to assess the rate of extracellular ferricyanide reduction that was presented as μmol
$(10^6 \, \text{cells})^{-1} \, \text{h}^{-1}$ (Nimer et al., 1998).
*2.8. Cell-driving pH drift experiment*
To obtain the pH compensation point, the cells were transferred to sealed glass vials
containing fresh medium (pH 8.2) with corresponding phosphate levels. The cell
concentration for all treatments was $5.0 \times 10^5 \, \text{mL}^{-1}$. The pH drift of the suspension was
monitored at $20^o$C and 200 μmol photons $\text{m}^{-2} \, \text{s}^{-1}$ light level. The pH compensation point was
obtained when there was no a further increase in pH.
*2.9. Statistical analysis*
Results were expressed as means of replicates ± standard deviation and data were
analyzed using the software SPSS v.21. The data from each treatment conformed to a normal
distribution (Shapiro-Wilk, $P > 0.05$) and the variances could be considered equal (Levene's
test, $P > 0.05$). Two-way ANOVAs were conducted to assess the effects of $CO_2$ and phosphate
on net photosynthetic rate, dark respiration rate, ratio of net photosynthetic rate to dark
respiration rate, rETR, Chl *a*, $K_{0.5}$, $CA_{ext}$, reduction rate of ferricyanide, and pH compensation
point. Least Significant Difference (LSD) was conducted for *post hoc* investigation. Repeated
measures ANOVAs were conducted to analyze the effects of DIC on net photosynthetic rate
and the effect of incubation time on media pH in a closed system. Bonferroni was conducted
for *post hoc* investigation as it is the best reliable *post hoc* test for repeated measures ANOVA
(Ennos, 2007). The threshold value for determining statistical significance was $P < 0.05$.

## 3. Results

*3.1. Effects of $CO_2$ and phosphate on photosynthetic and respiratory performances*

The net photosynthetic rate and dark respiration rate in *S. costatum* grown at various $CO_2$ and phosphate concentrations were first investigated (Fig. 1). $CO_2$ interacted with phosphate on net photosynthetic rate, with each factor having a main effect (Table 1 & Fig. 1a). *Post hoc* LSD comparison ($P = 0.05$) showed that 2.8 μmol $CO_2$ ~~LC~~ reduced net photosynthetic rate when the phosphate levels was below 4 μmol $L^{-1}$ but did not affect it at the higher phosphate levels. Under ~~AC~~the condition of 12.6 μmol $CO_2$, net photosynthetic rate increased with phosphate level and reached the plateau (100.51 ±9.59 μmol $O_2$ $(10^9 \text{ cells})^{-1} h^{-1}$) at 1 μmol $L^{-1}$ phosphate. Under the condition of 2.8 μmol $CO_2$ ~~LC~~, net photosynthetic rate also increased with phosphate level but did not hit the peak (101.46 ±9.19 μmol $O_2$ $(10^9 \text{ cells})^{-1} h^{-1}$) until 4 μmol $L^{-1}$ phosphate. In terms of dark respiration rate (Fig. 1b), phosphate had a main effect on it and it interacted with $CO_2$ (Table 1). Specifically, 2.8 μmol $CO_2$ ~~LC~~ increased dark respiration rate at 0.05 and 0.25 μmol $L^{-1}$ phosphate levels, but did not affect it when phosphate level was above 1 μmol $L^{-1}$ (LSD, $P < 0.05$). Regardless of $CO_2$ level, respiration rate increased with phosphate availability and stopped at 1 μmol $L^{-1}$.

The ratio of respiration to photosynthesis ranged from 0.23 to 0.40 (Fig. ~~2~~1c). Both $CO_2$ and phosphate had a main effect, and they interacted on the ratio of respiration to photosynthesis (Table 1). The level of 2.8 μmol $CO_2$ ~~LC~~ increased the ratio when phosphate was lower than 4 μmol $L^{-1}$ but did not affect it when phosphate levels were 4 or 10 μmol $L^{-1}$.

Both $CO_2$ and phosphate affected ~~r~~ETR and they also showed an interactive effect (Fig. ~~3~~

2 & Table 2). For instance, *post hoc* LSD comparison showed that 2.8 μmol $CO_2$LC did not
affect rETR at lower phosphate levels (0.05 and 0.25 μmol $L^{-1}$) but increased it at higher
phosphate levels (1–10 μmol $L^{-1}$). Regardless of $CO_2$ treatment, rETR increased with
phosphate level (0.05–4 μmol $L^{-1}$) but the highest phosphate concentration did not result in a
further increase in rETR (LSD, $P > 0.05$).
The content of Chl *a* was measured to investigate the effects of $CO_2$ and phosphate on
photosynthetic pigment in *S. costatum* (Fig. 43). Both $CO_2$ and phosphate affected the
synthesis of Chl *a* and they had an interactive effect (Table 2). *Post hoc* LSD comparison ($P =$
0.05) showed that 2.8 μmol $CO_2$LC did not affect Chl *a* at 0.05 or 0.25 μmol $L^{-1}$ phosphate
but stimulated Chl *a* synthesis at higher phosphate levels (1–10 μmol $L^{-1}$). Irrespective of $CO_2$
treatment, Chl *a* content increased with phosphate level and reached the plateau (0.19 ± 0.01
pg $cell^{-1}$ for 12.6 μmol $CO_2$AC and 0.23 ± 0.01 pg $cell^{-1}$ for 2.8 μmol $CO_2$LC) at 4 μmol $L^{-1}$
phosphate.
To assess the effects of $CO_2$ and phosphate on photosynthetic $CO_2$ affinity in *S. costatum*,
the net photosynthetic rates of cells exposure to seven levels of DIC were measured (Fig. 54).
After curve fitting, the values of $K_{0.5}$ for $CO_2$ were calculated (Fig. 65). $CO_2$ and phosphate
interplayed on $K_{0.5}$ and each had a main effect (Table 2). The level of 2.8 μmol $CO_2$LC did
not affect $K_{0.5}$ at the lowest phosphate level but reduced it at the other phosphate levels. Under
ACthe condition of 12.6 μmol $CO_2$, higher phosphate levels (0.25–4 μmol $L^{-1}$) reduced $K_{0.5}$
and the highest phosphate level led to a further decrease to 2.59 ± 0.29 μmol $kg^{-1}$ seawater
compared to the value of 4.00 ± 0.30 μmol $kg^{-1}$ seawater at  0.05 μmol $L^{-1}$ phosphate. The
pattern with phosphate under LCat 2.8 μmol $CO_2$ was the same as 12.6 μmol $CO_2$the AC.

*3.3. The effects of $CO_2$ and phosphate on inorganic carbon acquisition*
To investigate the potential mechanisms that cells overcame $CO_2$ limitation during algal
blooms, the activity of $CA_{ext}$, a CCM related enzyme, was estimated under various $CO_2$ and
phosphate conditions (Fig. ~~7a~~6a). Both $CO_2$ and phosphate had a main effect and they
interacted on $CA_{ext}$ activity (Table 3). *Post hoc* LSD comparison ($P = 0.05$) showed that 2.8
μmol $CO_2$ ~~LC~~ induced more $CA_{ext}$ activity under all phosphate conditions except for 0.05
μmol $L^{-1}$ levels, compared to 12.6 μmol $CO_2$ ~~AC~~. Under ~~AC~~the condition of 12.6 μmol $CO_2$,
$CA_{ext}$ activity increased (0.04–0.10 EU ($10^6$ cells)$^{-1}$) with phosphate level and stopped
increasing at 1 μmol $L^{-1}$ phosphate. Under the condition of 2.8 μmol $CO_2$ ~~LC~~, $CA_{ext}$ activity
also increased (0.04–0.35 EU ($10^6$ cells)$^{-1}$) with phosphate level but reached the peak at 4
μmol $L^{-1}$ phosphate. The redox activity of plasma membrane was also assessed to investigate
the factors that modulate $CA_{ext}$ activity (Fig. ~~7b~~6b). The pattern of redox activity of plasma
membrane under various $CO_2$ and phosphate conditions was the same as that of $CA_{ext}$ activity.
That is, $CO_2$ and phosphate had an interactive effect on redox activity of plasma membrane,
each having a main effect (Table 3).
To test cells' tolerance to high pH and obtain pH compensation points in *S. costatum*
grown under various $CO_2$ and phosphate levels, changes of media pH in a closed system were
monitored (Fig. ~~8~~7). The media pH under all phosphate conditions increased with incubation
time (Table 4). Specifically speaking, there was a steep increase in pH during the first three
hours, afterwards the increase became slower and it reached a plateau in six hours (Bonferroni,
$P < 0.05$). Phosphate had an interactive effect with incubation time (Table 4). For instance,
there was no significant difference in media pH among phosphate levels during first two
hours of incubation but then divergence occurred and they stopped at different points.
Two-way ANOVA analysis showed that $CO_2$ treatment did not affect pH compensation point
but phosphate had a main effect (Table 3-). Under each $CO_2$ treatment, pH compensation point
increased with phosphate level, with lowest of 9.03 ± 0.03 at 0.05 μmol L$^{-1}$ and highest of
9.36 ± 0.04 at 10 μmol L$^{-1}$ phosphate.
**4.   Discussion**
*4.1. Photosynthetic performances under various $CO_2$ and phosphate conditions*
The lower $CO_2$ availability reduced the net photosynthetic rate of *S. costatum* grown at
the lower phosphate levels in the present study. However, Nimer *et al.* (1998) demonstrated
that the increase in pH (8.3−9.5) did not reduce photosynthetic $CO_2$ fixation of *S. costatum*
and Chen and Gao (2004) reported that a higher pH (8.7) even stimulated the photosynthetic
rate of *S. costatum* compared to the control (pH 8.2). The divergence between our and the
previous studies may be due to different nutrient supply. Both Nimer *et al.* (1998) and Chen
and Gao (2004) used f/2 media to grow algae. The phosphate concentration in f/2 media is
~36 μmol L$^{-1}$, which is replete for physiological activities in *S. costatum*. *Skeletonema*
*costatum* grown at higher phosphate levels (4 and 10 μmol L$^{-1}$) also showed similar
photosynthetic rates for the lower and higher $CO_2$ treatments. Our finding combined with the
previous studies indicates phosphorus plays an important role in dealing with low $CO_2$
availability for photosynthesis in *S. costatum*.
Different from net photosynthetic rate, 2.8 μmol $CO_2$~~LC~~ did not affect rETR at lower
phosphate levels (0.05 and 0.25 μmol L$^{-1}$) and stimulated it at higher phosphate levels (1−10
μmol L$^{-1}$). This interactive effect of $CO_2$ and phosphate may be due to their effects on Chl *a*.

The level of 2.8 μmol $CO_2$LC induced more synthesis of Chl *a* at higher phosphate levels (1−10 μmol L$^{-1}$). This induction of lower $CO_2$LC on photosynthetic pigment is also reported in green algae (Gao *et al.*, 2016). More energy is required under LC lower $CO_2$ to address the more severe $CO_2$ limitation and thus more Chl *a* are synthesized to capture more light energy, particularly when phosphate was replete. Although P is not an integral component for chlorophyll, it plays an important role in cell energetics through high-energy phosphate bonds, i.e. ATP, which could support chlorophyll synthesis. The stimulating effect of P enrichment on photosynthetic pigment is also found in green alga *Dunaliella tertiolecta* (Geider *et al.*, 1998) and brown alga *Sargassum muticum* (Xu *et al.*, 2017). The increased photosynthetic pigment in *S. costatum* could partially explain the increased rETR and photosynthetic rate under the higher P conditions.

*4.6.  Ratio of respiration to photosynthesis*

The ratio of respiration to photosynthesis in algae indicates carbon balance in cells and carbon flux in marine ecosystems as well (Zou & Gao, 2013).The level of 2.8 μmol $CO_2$LC increased this ratio in *S. costatum* grown at the lower P conditions but did not affect it under the higher P conditions, indicating that P enrichment can offset the carbon loss caused by carbon limitation. To cope with $CO_2$ limitation, cells might have to obtain energy from dark respiration under lower P conditions as it seems infeasible to acquire energy from the low rETR, which led to the increased dark respiration. However, 2.8 μmol $CO_2$LC induced higher rETR under P replete conditions and energy used for inorganic carbon ($CO_2$ and $HCO_3^-$) acquisition could be from the increased rETR. Therefore, additional dark respiration was not triggered, avoiding carbon loss. Most studies regarding the effect of $CO_2$ on ratio of

respiration to photosynthesis focus on higher plants (Gifford, 1995; Ziska & Bunce, 1998;
Cheng *et al.*, 2010; Smith & Dukes, 2013), little is known on phytoplankton. Our study
suggests that $CO_2$ limitation may lead to carbon loss in phytoplankton but P enrichment could
alter this trend, regulating carbon balance in phytoplankton and thus their capacity in carbon
sequestration.
*4.3. Inorganic carbon acquisition under $CO_2$ limitation and phosphate enrichment*
Decreased $CO_2$ can usually induce higher inorganic carbon affinity in algae (Raven *et al.*,
2012; Wu *et al.*, 2012; Raven *et al.*, 2017; Xu *et al.*, 2017). In the present study, the lower
$CO_2$ did increase inorganic carbon affinity when P level was higher than 0.25 μmol $L^{-1}$ but did
not affect it when P was 0.05 μmol $L^{-1}$, indicating the important role of P in regulating cells'
CCMs in response to environmental $CO_2$ changes. The level of 2.8 μmol $CO_2$ LC induced
larger CA activity when P was above 0.25 μmol $L^{-1}$ but did not increase it at 0.05 μmol $L^{-1}$ of
P, which could explain the interactive effect of P and $CO_2$ on inorganic carbon affinity as CA
can accelerate the equilibrium between $HCO_3^-$ and $CO_2$ and increase inorganic carbon affinity.
Regardless of $CO_2$, P enrichment alone increased CA activity and inorganic carbon affinity. P
enrichment may stimulate the synthesis of CA by supplying required ATP. In addition, P
enrichment increased the redox activity of plasma membrane in this study. It has been
proposed that redox activity of plasma membrane could induce extracellular CA activity via
protonation extrusion of its active center (Nimer *et al.*, 1998). Our result that the pattern of
CA is exactly the same as that of redox activity of plasma membrane shows a compelling
correlation between CA and redox activity of plasma membrane. The stimulating effect of P
on redox activity of plasma membrane may be due to its effect on rETR. The increased rETR
could generate excess reducing equivalents, particularly under $CO_2$ limiting conditions. These
excess reducing equivalents would be transported from the chloroplast into the cytosol (Heber,
1974), supporting the redox chain in the plasma membrane (Rubinstein & Luster, 1993;
Nimer *et al.*, 1999) and triggering CA activity.
*4.4. Direct $HCO_3^-$ utilization due to phosphate enrichment*
A pH compensation point over 9.2 has been considered a sign of direct $HCO_3^-$ use for
algae (Axelsson & Uusitalo, 1988) as the $CO_2$ concentration is nearly zero at pH above 9.2.
This criterion has been justified based on ~~the~~ experiments for both micro and macro-algae.
For instance, the marine diatom *Phaeodactylum tricornutum*, with a strong capacity for direct
$HCO_3^-$ utilization, has a higher pH compensation point of 10.3 (Chen *et al.*, 2006). In contrast,
the red macroalgae, *Lomentaria articulata* and *Phycodrys rubens* that cannot utilize $HCO_3^-$
directly, and whose photosynthesis only depends on $CO_2$ diffusion, have pH compensation
points of less than 9.2 (Maberly, 1990). In terms of *S. costatum*, it has been reported to have a
pH compensation point of 9.12, indicating a very weak capacity in direct $HCO_3^-$ utilization
(Chen & Gao, 2004). Our study demonstrates that the pH compensation point of *S. costatum*
varies with the availability of P. It is lower than 9.2 under P limiting conditions but higher
than 9.2 under P replete conditions, suggesting that the capacity of direct $HCO_3^-$ utilization is
regulated by P availability. Contrary to $CO_2$ passive diffusion, the direct use of $HCO_3^-$
depends on positive transport that requires energy (Hopkinson & Morel, 2011). P enrichment
increased ~~r~~ETR in the present study and the ATP produced during the process of electron
transport could be used to support $HCO_3^-$ positive transport. In addition, the increased
respiration at higher P levels can also generate ATP to help $HCO_3^-$ positive transport. Our
study indicates that P enrichment could trigger $HCO_3^-$ direct utilization and hence increase
inorganic acquisition capacity of *S. costatum* to cope with $CO_2$ limitation.
*4.5. CCMs and red tides*
In~~With~~ the development of red tides, the pH in seawater can~~ould~~ be very high along with
extremely low $CO_2$ availability due to intensive photosynthesis (Hansen, 2002; Hinga, 2002).
For instance, pH level in the surface waters of the eutrophic Mariager Fjord, Denmark, is
often above 9 during dinoflagellate blooms (Hansen, 2002). Diatoms are the casautive species
for red tides and *S. costatum* could outcompete other bloom algae (dinoflagellates
*Prorocentrum minimum* and *Alexandrium tamarense*) under nutrient replete conditions (Hu *et*
*al.*, 2011). However, the potential mechanisms are poorly understood. Our study demonstrates
*S. costatum* has multiple CCMs to cope with $CO_2$ limitation and the operation of CCMs is
regulated by P availability. The CCMs of *S. costatum* are hampered under P limiting
conditions and only function when P is replete. This finding may explain why ~~Therefore, P~~
~~enrichment would be critical for *S. costatum* to~~ diatoms could overcome carbon limitation
~~during algal bloom and to~~and dominate red tides when P is replete and as well as the shift
from diatoms to dinoflagellates when P is limiting (Mackey et al., 2012).
**5. Conclusions**
The present study investigated the role of P in regulating inorganic carbon acquisition and
$CO_2$ concentrating mechanisms in diatoms for the first time. The intensive photosynthesis and
quick growth during algal blooms usually result in noticeable increase of pH and decrease of
$CO_2$. Our study demonstrates that P enrichment could induce activity of extracellular carbonic
anhydrase and direct utilization of $HCO_3^-$ in *S. costatum* to help overcome ~~the~~ $CO_2$ limitation,
as well as increasing photosynthetic pigment content and rETR to provide required energy.
This study provides important insight into the connection of phosphorus and carbon
acquisition in diatoms and the mechanisms that help *S. costatum* dominates algal blooms.
**Author contribution**
JX and GG designed the experiments, and GG, JY, JF and XZ carried them out. GG
prepared the manuscript with contributions from all co-authors.
**Acknowledgements**
This work was supported by National Natural Science Foundation of China (No.
41376156&40976078), Natural Science Fund of Guangdong Province (No. S2012010009853),
the China Postdoctoral Science Foundation (2018T110463&2017M620270), Jiangsu Planned
Projects for Postdoctoral Research Funds (1701003A), Science Foundation of Huaihai
Institute of Technology (Z2016007), and
Foundation for High-level Talents in Higher Education of Guangdong.

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

Table 1 Two-way analysis of variance for the effects of $CO_2$ and phosphate on net
photosynthetic rate, dark respiration rate and ratio of respiration to photosynthesis of *S.*
*costatum*. $CO_2$*phosphate means the interactive effect of $CO_2$ and phosphate, df means degree
of freedom, F means the value of F statistic, and Sig. means p-value.

| Source | Net photosynthetic rate | | | Dark respiration rate | | | Ratio of respiration to photosynthesis | | |
|---|---|---|---|---|---|---|---|---|---|
| | df | F | Sig. | df | F | Sig. | df | F | Sig. |
| $CO_2$ | 1 | 11.286 | 0.003 | 1 | 1.262 | 0.275 | 1 | 32.443 | <0.001 |
| Phosphate | 4 | 157.925 | <0.001 | 4 | 169.050 | <0.001 | 4 | 7.081 | 0.001 |
| $CO_2$*phosphate | 4 | 3.662 | 0.021 | 4 | 3.226 | 0.034 | 4 | 8.299 | <0.001 |
| Error | 20 | | | 20 | | | 20 | | |

Table 2 Two-way analysis of variance for the effects of $CO_2$ and phosphate on relative
electron transport rate (rETR), Chl $a$, and $CO_2$ level required to give half of DIC-saturated
maximum rate of photosynthetic $O_2$ evolution ($K_{0.5}$) of *S. costatum*. $CO_2$*phosphate means the
interactive effect of $CO_2$ and phosphate, df means degree of freedom, F means the value of F
statistic, and Sig. means p-value.

| Source | rETR | | | Chl $a$ | | | $K_{0.5}$ | | |
|---|---|---|---|---|---|---|---|---|---|
| | df | F | Sig. | df | F | Sig. | df | F | Sig. |
| $CO_2$ | 1 | 28.717 | <0.001 | 1 | 32.963 | <0.001 | 1 | 96.182 | <0.001 |
| Phosphate | 4 | 127.860 | <0.001 | 4 | 92.045 | <0.001 | 4 | 40.497 | <0.001 |
| $CO_2$*phosphate | 4 | 3.296 | 0.031 | 4 | 3.871 | 0.017 | 4 | 3.821 | 0.018 |
| Error | 20 | | | 20 | | | 20 | | |

Table 3 Two-way analysis of variance for the effects of $CO_2$ and phosphate on $CA_{ext}$ activity,
redox activity of plasma membrane and pH compensation point of *S. costatum*.
$CO_2$*phosphate means the interactive effect of $CO_2$ and phosphate, df means degree of
freedom, F means the value of F statistic, and Sig. means p-value.

| Source | $CA_{ext}$ activity | | | redox activity of plasma membrane | | | pH compensation point | | |
|---|---|---|---|---|---|---|---|---|---|
| | df | F | Sig. | df | F | Sig. | df | F | Sig. |
| $CO_2$ | 1 | 569.585 | <0.001 | 1 | 937.963 | <0.001 | 1 | 0.056 | 0.816 |
| Phosphate | 4 | 176.392 | <0.001 | 4 | 276.362 | <0.001 | 4 | 226.196 | <0.001 |
| $CO_2$*phosphate | 4 | 87.380 | <0.001 | 4 | 137.050 | <0.001 | 4 | 0.040 | 0.997 |
| Error | 20 | | | 20 | | | 20 | | |

Table 4 Repeated measures analysis of variance for the effects of $CO_2$ and phosphate on pH
change during 10 hours of incubation. Time*$CO_2$ means the interactive effect of incubation
time and $CO_2$, Time*phosphate means the interactive effect of incubation time and phosphate,
Time*$CO_2$*phosphate means the interactive effect of incubation time, $CO_2$ and phosphate,
df means degree of freedom, F means the value of F statistic, and Sig. means p-value.

| Source | Type III Sum of Squares | df | Mean Square | F | Sig. |
|---|---|---|---|---|---|
| Time | 40.766 | 10 | 4.077 | 8737.941 | <0.001 |
| Time*$CO_2$ | 0.003 | 10 | <0.001 | 0.569 | 0.838 |
| Time*phosphate | 0.886 | 40 | 0.022 | 47.496 | <0.001 |
| Time*$CO_2$*phosphate | 0.002 | 40 | <0.001 | 0.112 | 1.000 |
| Error | 0.093 | 200 | <0.001 | | |

**Figure legends**

**Fig. 1.** Net photosynthetic rate (a) ~~and~~, dark respiration rate (b) and ratio of respiration rate to net photosynthetic rate (c) in *S. costatum* grown at various phosphate concentrations after 12.6~~ambient (AC)~~ and 2.8 μmol $CO_2$ ~~low $CO_2$ (LC)~~ treatments. The error bars indicate the standard deviations (n = 3). ~~Different letters represent the significant difference ($P < 0.05$) among phosphate concentrations (capital for AC, lower case for LC). Horizontal lines represent significant difference ($P < 0.05$) between $CO_2$ treatments.~~

~~**Fig. 2.** Ratio of respiration rate to net photosynthetic rate in *S. costatum* grown at various phosphate concentrations after ambient (AC) and low $CO_2$ (LC) treatments. The error bars indicate the standard deviations (n = 3). Different letters represent the significant difference ($P < 0.05$) among phosphate concentrations (capital for AC, lower case for LC). Horizontal lines represent significant difference ($P < 0.05$) between $CO_2$ treatments.~~

**Fig. ~~3~~2.** Relative electron transport rate (rETR) in *S. costatum* grown at various phosphate concentrations after 12.6 and 2.8 μmol $CO_2$~~ambient (AC) and low $CO_2$ (LC)~~ treatments. The error bars indicate the standard deviations (n = 3). ~~Different letters represent the significant difference ($P < 0.05$) among phosphate concentrations (Capital for AC lower case for LC). Horizontal lines represent significant difference ($P < 0.05$) between $CO_2$ treatments.~~

**Fig. ~~4~~3.** Photosynthetic Chl *a* content in *S. costatum* grown at various phosphate concentrations after 12.6 and 2.8 μmol $CO_2$~~ambient (AC) and low $CO_2$ (LC)~~ treatments. The error bars indicate the standard deviations (n = 3). ~~Different letters represent the significant difference ($P < 0.05$) among phosphate concentrations (capital for AC, lower case for LC). Horizontal lines represent significant difference ($P < 0.05$) between $CO_2$ treatments.~~

**Fig. ~~5~~4.** Net photosynthetic rate as a function of DIC for *S. costatum* grown at various
phosphate concentrations after 12.6 ~~ambient~~ (a) and ~~–~~ 2.8 $\mu$mol $CO_2$~~low CO₂~~ (b) treatments.
The error bars indicate the standard deviations (n = 3).
**Fig. ~~6~~5.** Half saturation constant ($K_{0.5}$) for $CO_2$ in *S. costatum* grown at at various phosphate
concentrations after 12.6 and 2.8 $\mu$mol $CO_2$~~ambient (AC) and low CO₂ (LC)~~ treatments. The
error bars indicate the standard deviations (n = 3). ~~Different letters represent the significant~~
~~difference ($P < 0.05$) among phosphate concentrations (capital for AC, lower case for LC).~~
~~Horizontal lines represent significant difference ($P < 0.05$) between CO₂ treatments.~~
**Fig. ~~7~~6.** $CA_{ext}$ activity (a) and reduction rate of ferricyanide (b) in *S. costatum* grown at
various phosphate concentrations after 12.6 and 2.8 $\mu$mol $CO_2$~~ambient (AC) and low CO₂ (LC)~~
treatments. The error bars indicate the standard deviations (n = 3). ~~Different letters represent~~
~~the significant difference ($P < 0.05$) among phosphate concentrations (capital for AC, lower~~
~~case for LC). Horizontal lines represent significant difference ($P < 0.05$) between CO₂~~
~~treatments.~~
**Fig. ~~8~~7.** Changes of pH in a closed system caused by photosynthesis of *S. costatum* grown at
various phosphate concentrations after 12.6 and 2.8 $\mu$mol $CO_2$~~ambient (AC) and low CO₂ (LC)~~
treatments. The error bars indicate the standard deviations (n = 3).

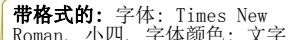

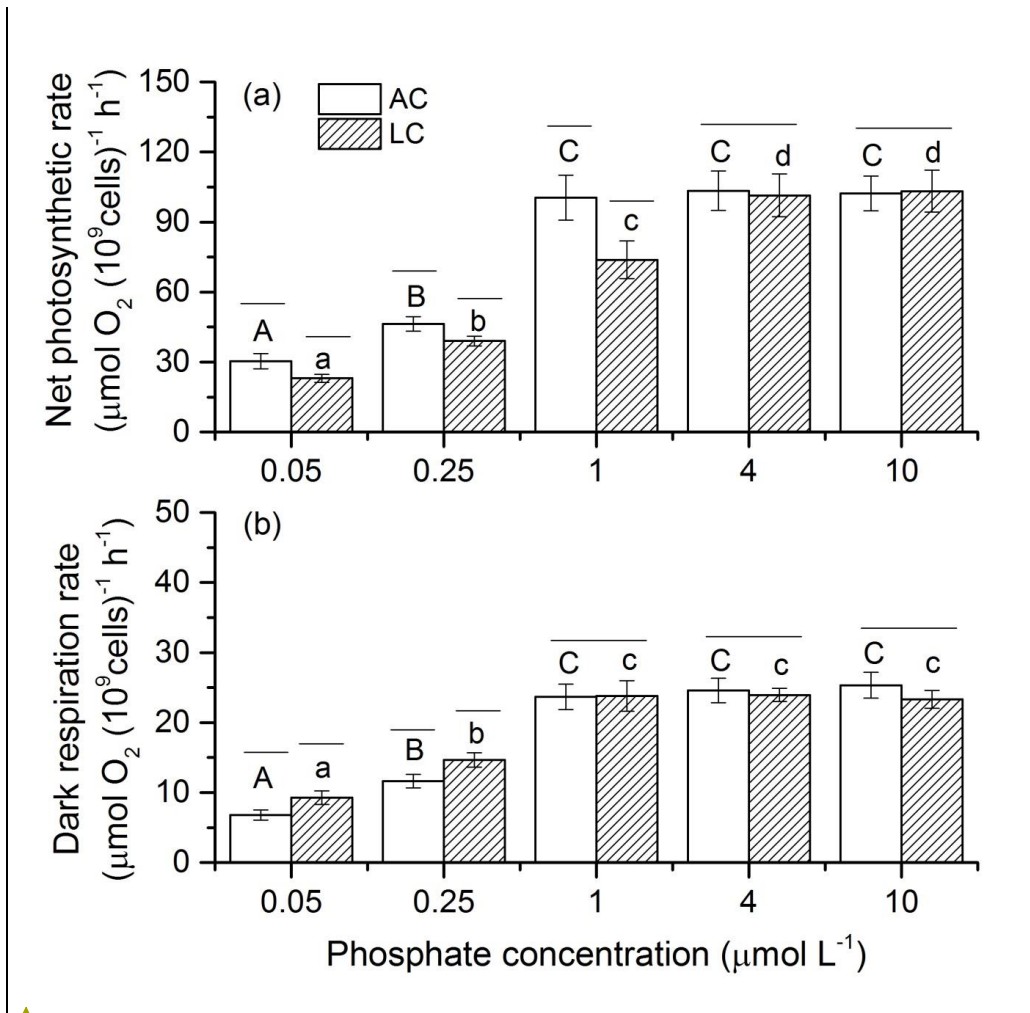


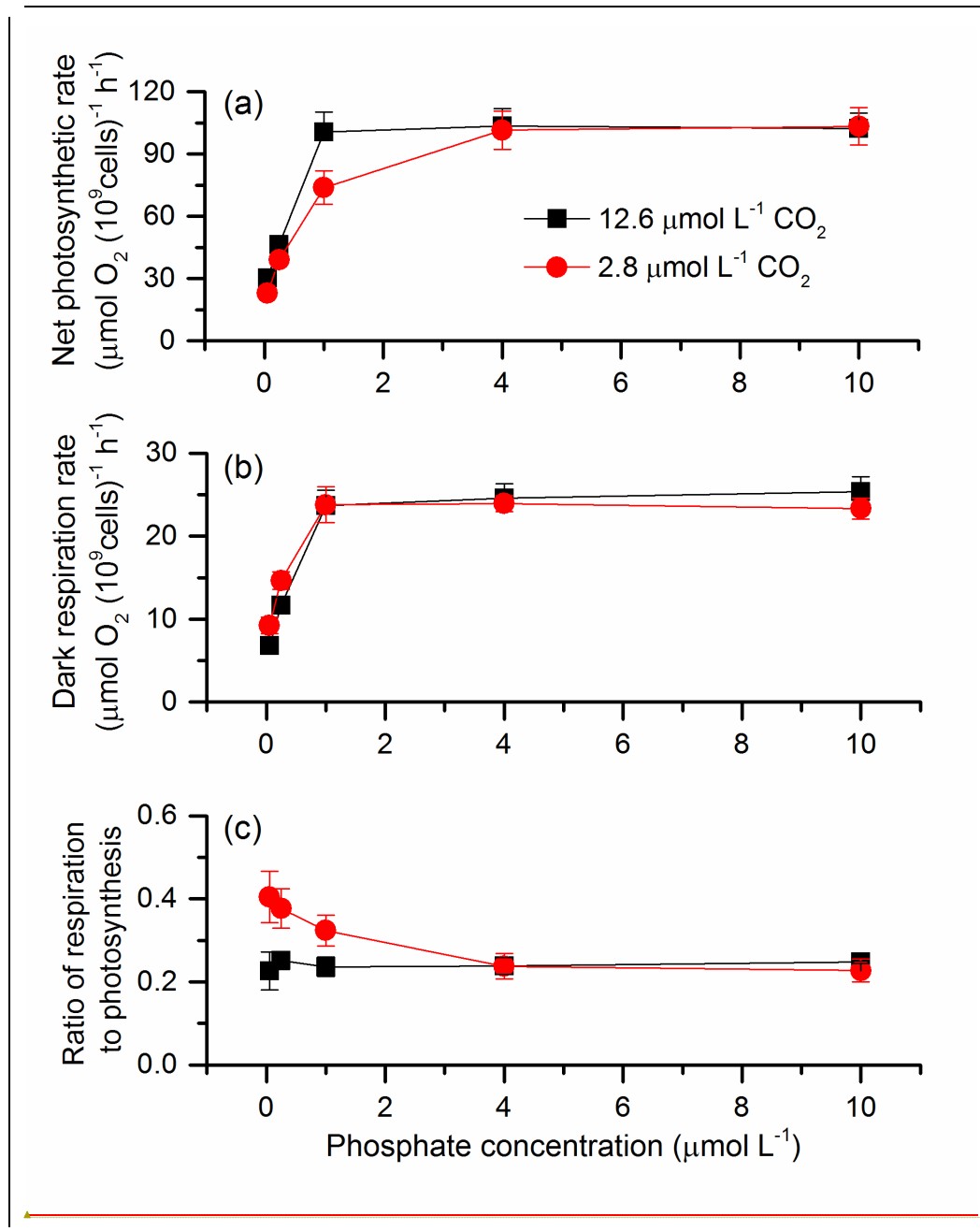



**Fig. 1**

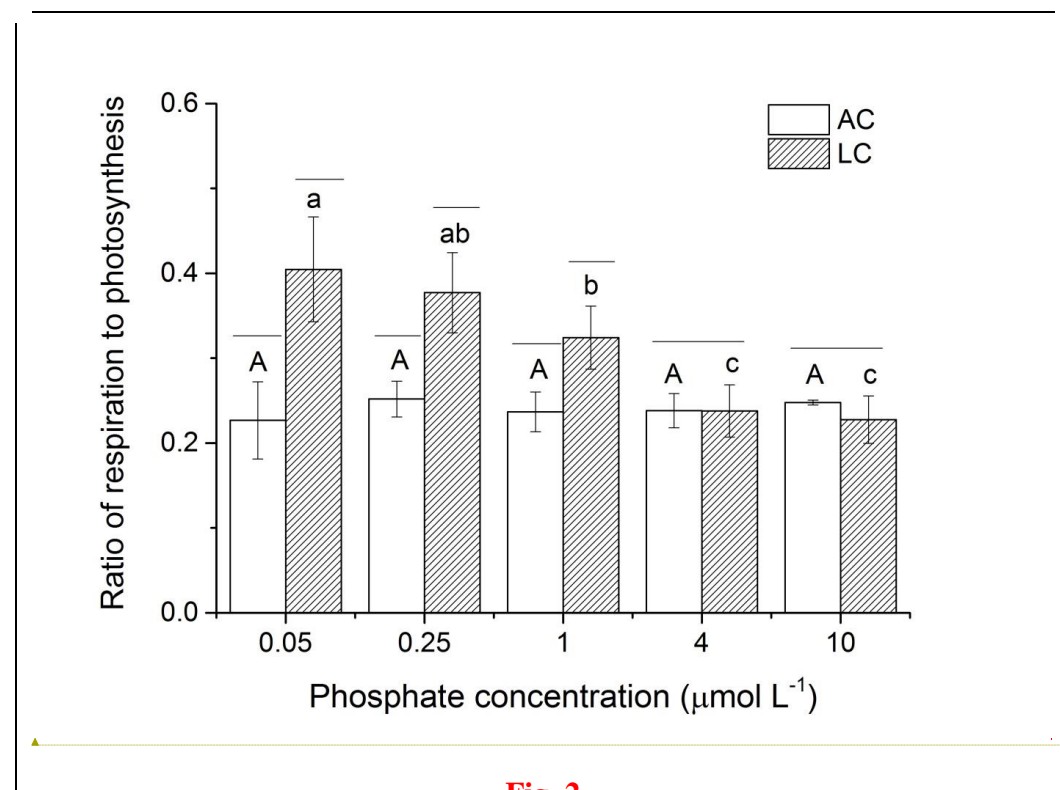


**Fig. 2**

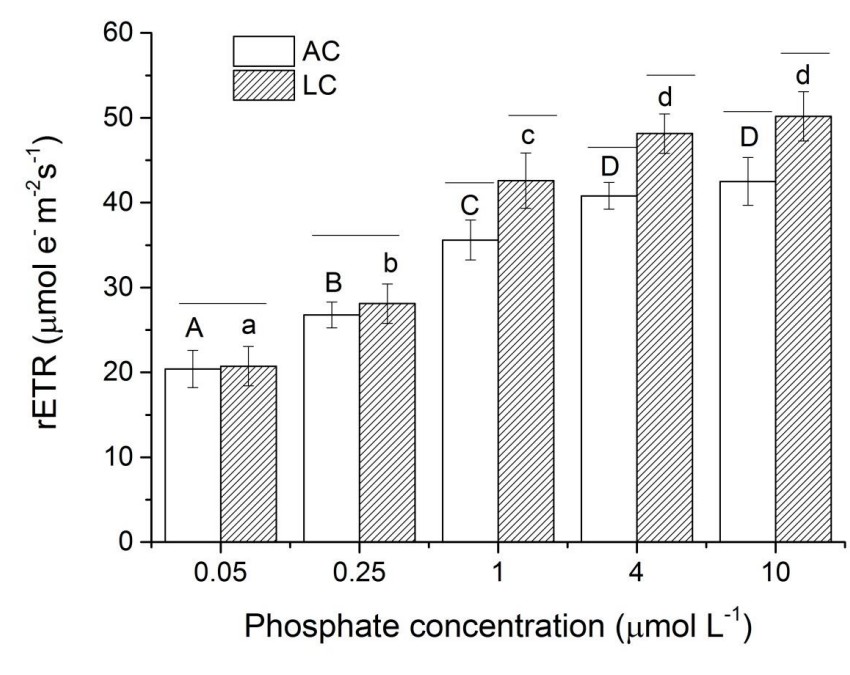


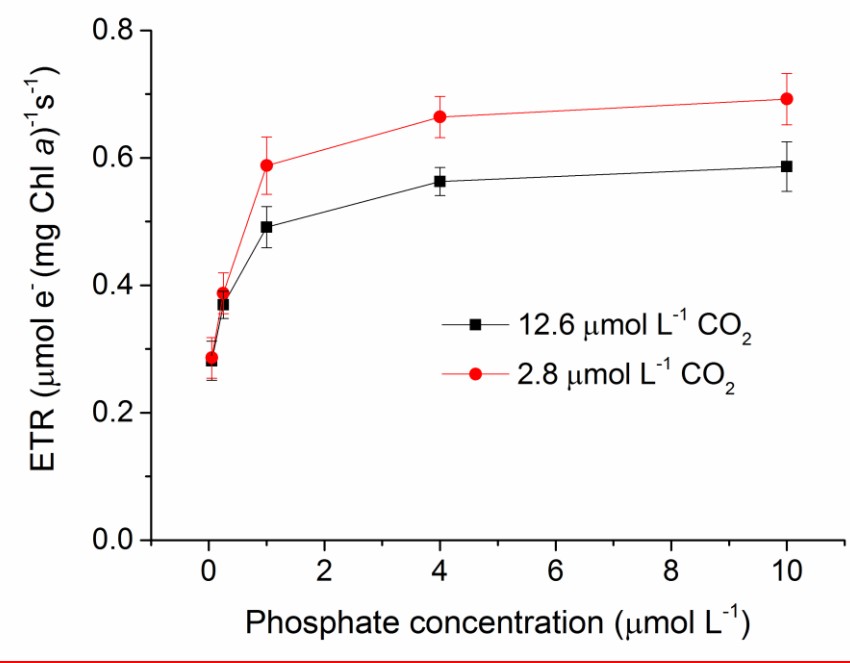

**Fig. ~~3~~2**

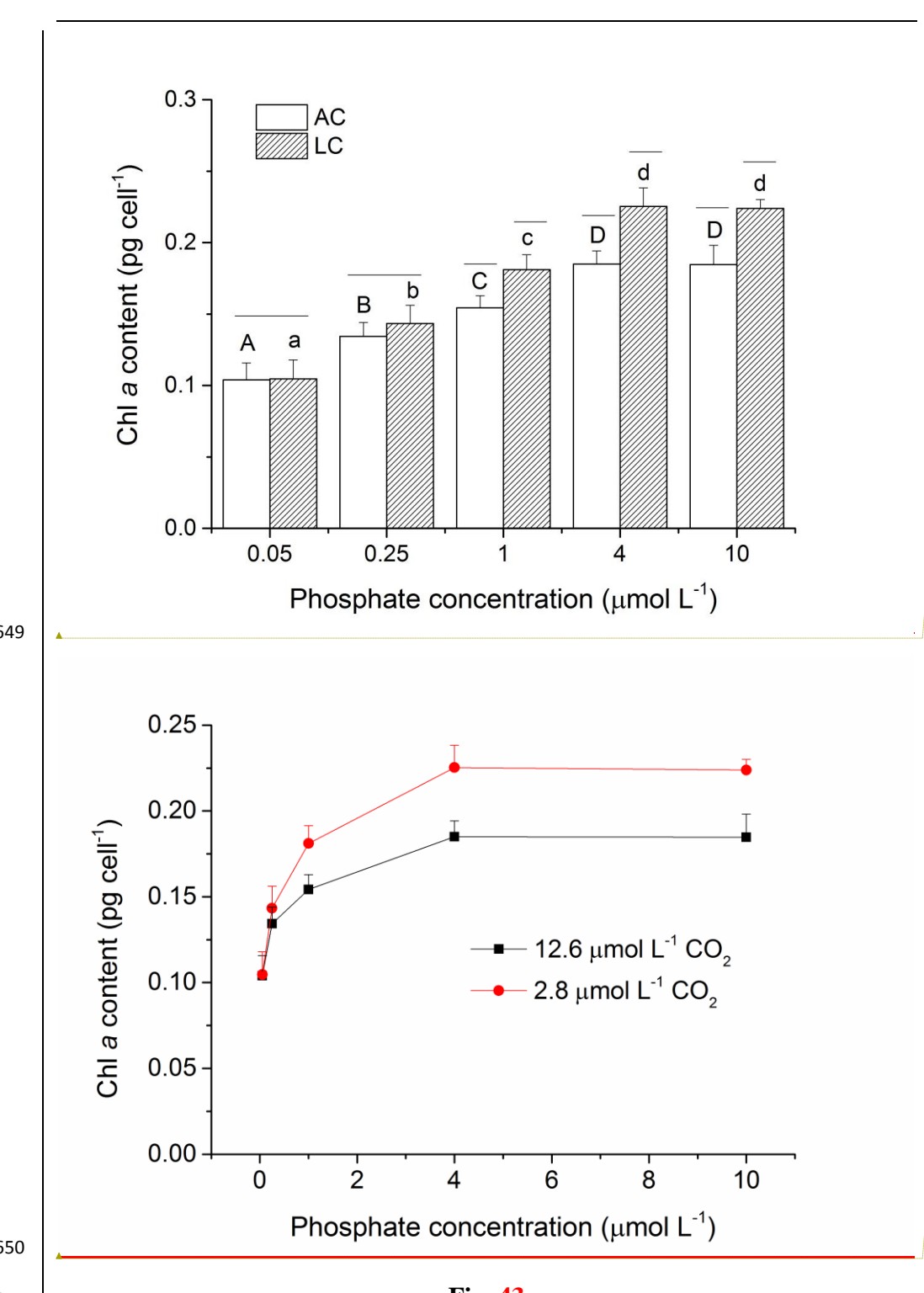

**Fig. 43**

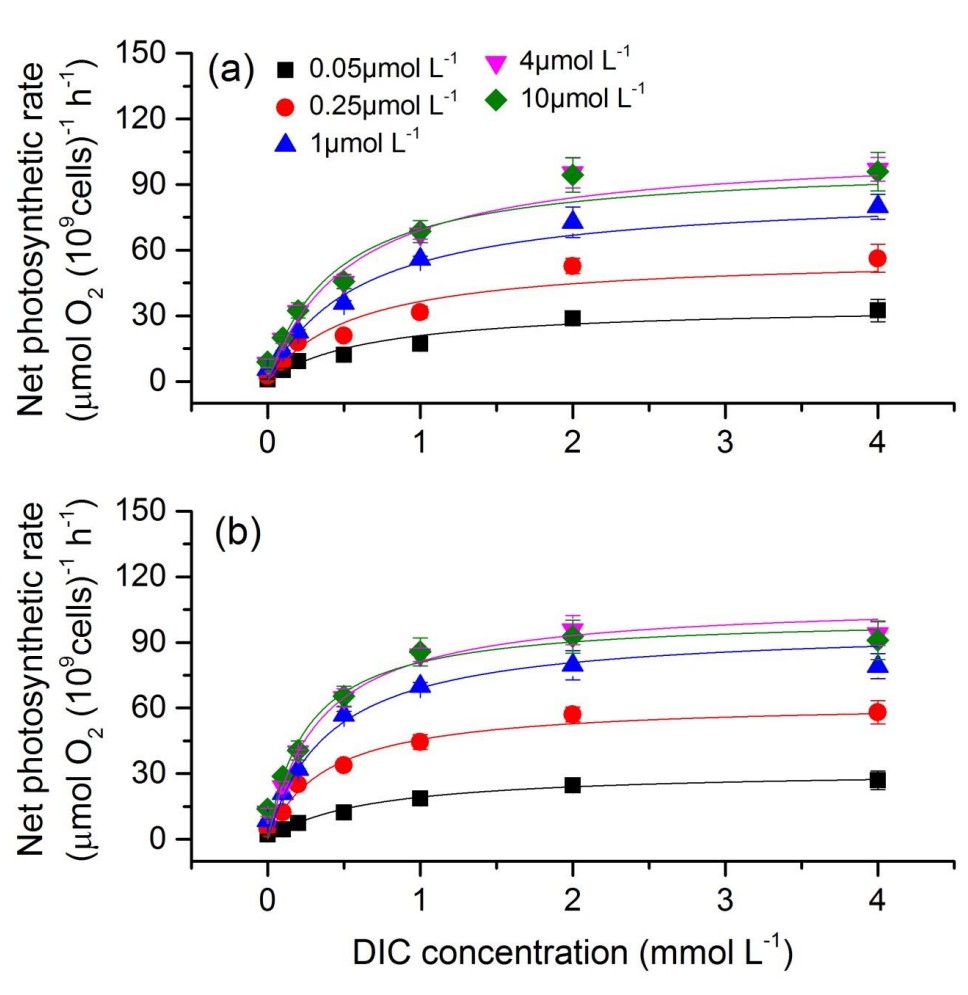


653                                             **Fig. 54**

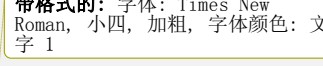

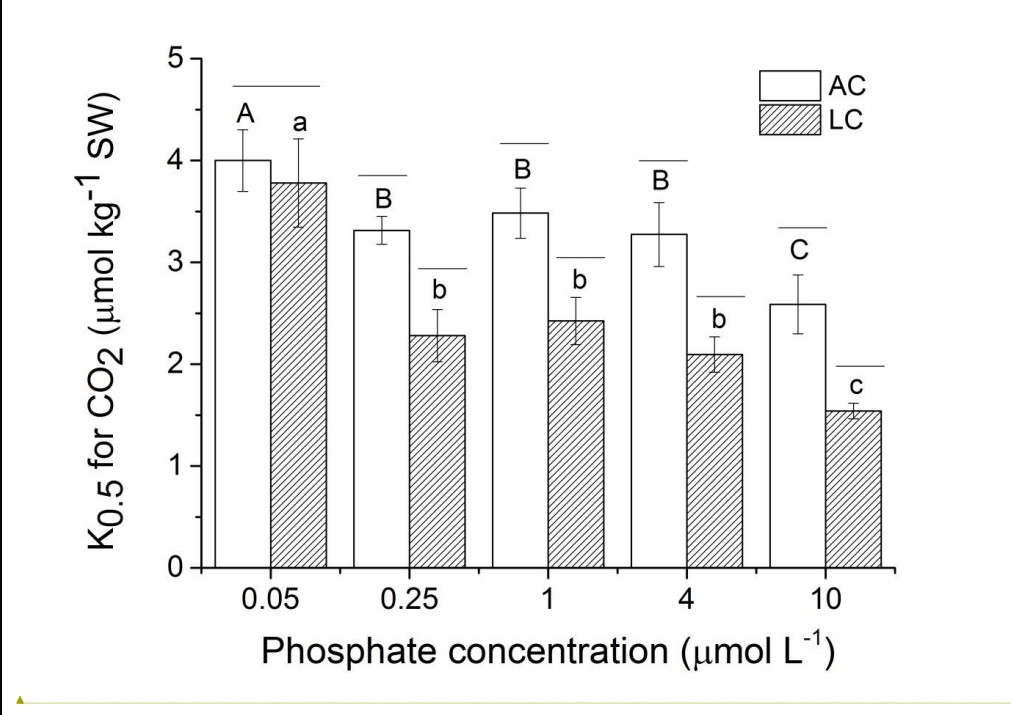


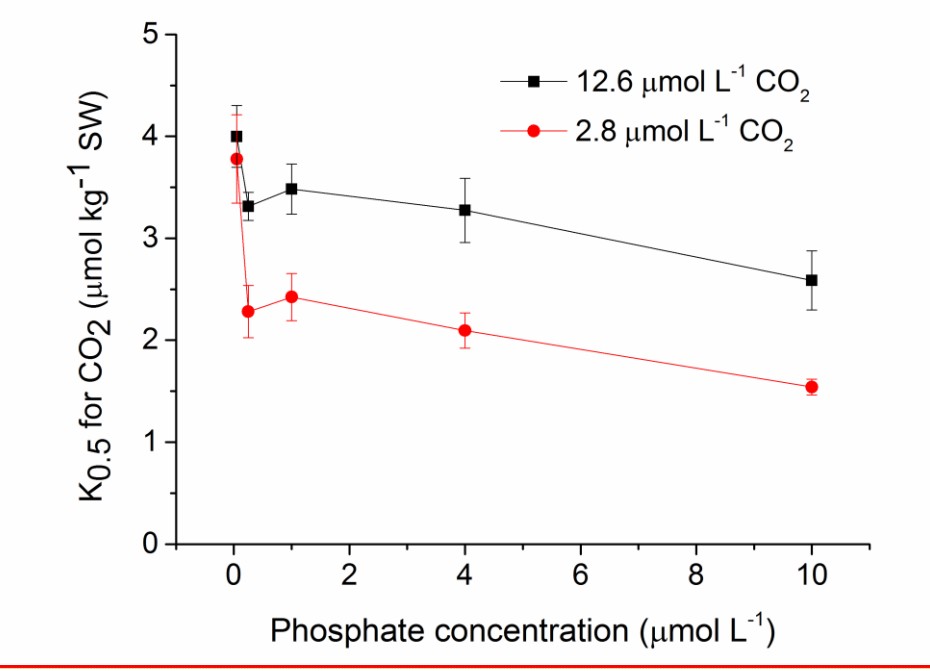


**Fig. 65**

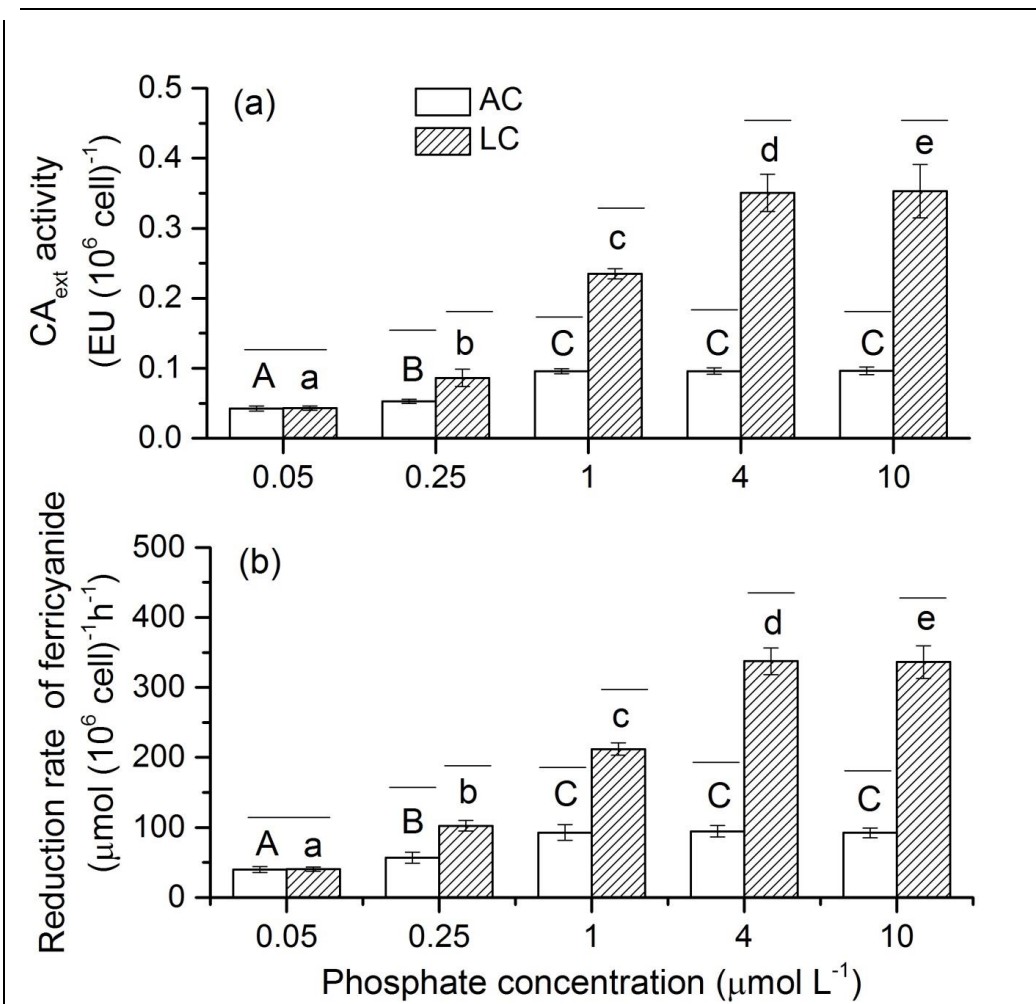

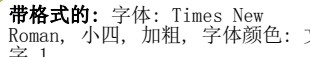

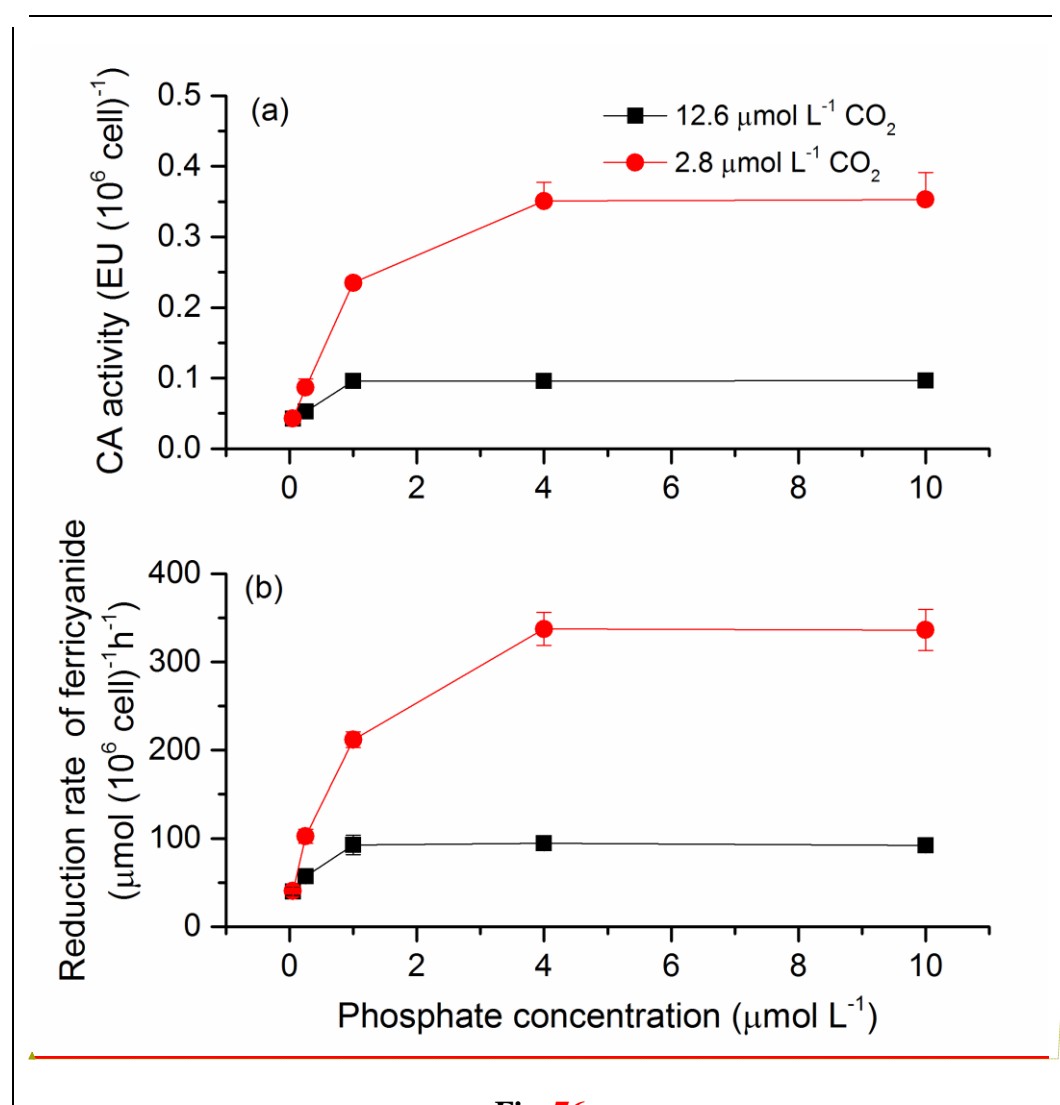



**Fig. 76**

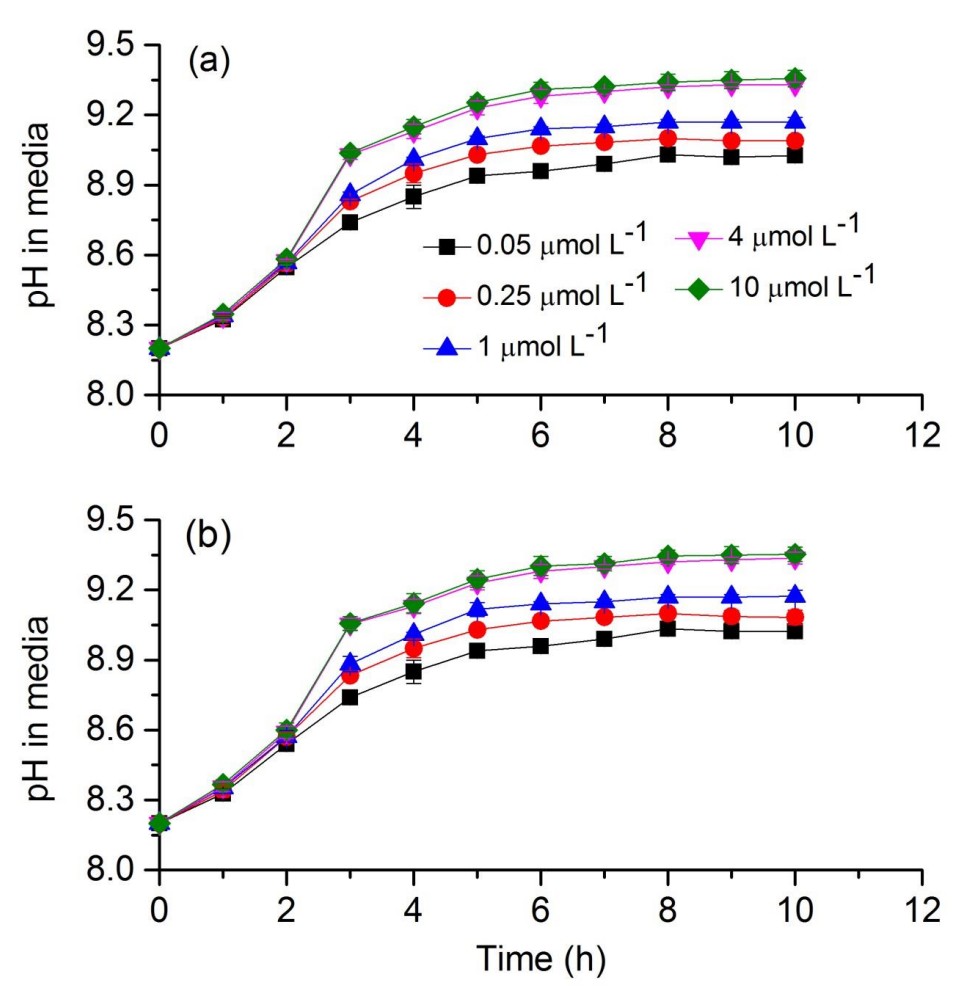

**Fig. 87**