# Peer review of "Regulation of inorganic carbon acquisition in a red tide alga (*Skeletonema costatum*): the importance of phosphorus availability"

_Biogeosciences, 2017_

## Referee Comment (RC1) · C.W. Hunt (Referee) · 14 Feb 2018

Review of Gao et al. "Regulation of inorganic acquisition in a red tide alga (Skeltonema costatum): the importance of phosphorus availability

The authors present a manuscript detailing culture studies of a common algae species under varying CO2 and phosphorus concentrations. Algal blooms can draw down dissolved CO2 to very low levels, and some species have developed mechanisms to compensate for decreased CO2 availability. Some of these mechanisms may be influenced by the presence or absence of bio-available phosphorus, leading to the study design of varying phosphorus levels across test populations of two CO2 levels. While there

is probably a compelling study underlying this manuscript, in my opinion there are too many flaws as presented to encourage publication. I feel compelled to point out here that my background is in seawater carbonate chemistry, and my knowledge of cellular biochemical processes is limited. However, based solely on the description of inorganic carbon system measurements I would advise rejecting this manuscript until serious revisions have been done. I will present some major comments below, followed by more minor concerns.

MAJOR COMMENTS -The Materials and Methods section, in particular the sections pertaining to pH and alkalinity measurements, is totally inadequate. Is the pH system an electrode-type system? What pH scale are measurements presented in? How was the pH system calibrated? How were alkalinity titrations performed? No information is presented. These questions are especially critical in the calculation of DIC from pH and alkalinity (P7L129-131), which is very sensitive to relatively minor changes in pH and alkalinity. With no information about the quality of pH and alkalinity measurements, the results of this analysis are impossible to interpret.

-Besides using the barely-described pH system, how did the authors know the CO2 levels of their cultures? Also, adding phosphate to the cultures, at concentrations ranging from 0.5-10 $\mu$mol/kg adds a potent buffering agent, as monosodium phosphate has a pKa around 7. How did the authors alter or maintain the pH in the cultures? Were the cultures open to the ambient atmosphere?

-As previously mentioned, my knowledge of some of the biochemical processes presented here is minimal, and the Introduction did little to help readers like myself. There seem to be connections between plasma membrane redox activity, CAext, rETR, but the manuscript does not explain them. Some terms (i.e. rETR) are presented with no explanation or definition. Thus the reason for some of the analyses presented was unclear to me. What did measuring the chlorophyll fluorescence inform? The cultures were initiated at the same cell density, but surely the cell density varied between cultures after the treatment period- how was this accounted for?

-In the Results section the authors present their statistical findings in the form $F_{(1,20)}=XX$ or $F_{(4,20)}=XX$. I'm assuming these are the results of the two-way ANOVA test mentioned in the 2.8 Statistical Analysis section, but no explanation is given as to what is signified. Are the numbers in parentheses indicating degrees of freedom? What is the threshold for significance?

-The Results section is extremely repetitive. Much of the information presented could be more effectively summarized in a table.

MINOR COMMENTS

-The English usage in much of the manuscript could be improved. I will note some points below.

-Define rETR in the Abstract (P2L10)

-P2L3 and throughout: define the abbreviations when first used: CO2, DIC, HCO3-, rETR

-P2L16 change "is" to "was"

-P3L26 change to "the marine biological"

-P3L31 need a different word than compelling

-P3L33 and throughout: don't finish sentences with "etc".

-P3L34 change "could" to "can".

-P3L36 misselled "dominate"

-P3L37 P3L40-41 How is RUBISCO important? What is it, an enzyme?

-P4L45-48: the carbon concentrating mechanisms are named but not explained. A reader like myself has no way to know what "multiple carbon anhydrase, assumed C4-type pathway" represents

-P4L54 define CCMs (CO2 concentrating mechanisms, right?)

-P4L57: keep consistent units between discussions of CO2 or pCO2. Discuss either CO2 concentration or partial pressure. Reader has no way to compare 5 $\mu$mol/L CO2 to a pCO2 of 1800 $\mu$atm.

-P5L69: cite the refernces yourself, don't refer to references therein

-P5L70: what is the relationship?

-P5L75-76: remove "the capacity of"

-P5L78-79: all these mechanisms/pathways! How do they interrelate?

-P5L82: as CO2 is removed by diatom growth, the inorganic carbon equilibria will shift to convert HCO32- to CO2. How do the kinetics of this potentially affect this study?

-P6L105: what does "algae after in light" mean?

-P6L108: change to "photosynthetic and respiration rates"

-P7L114-116: why measure photosynthesis for 5 minutes but respiration for 10?

-P7L126: what is "Ci-saturated maximum rate"?

-P8L149: by "samples" do you mean the diatom cells?

-P9L156: what is the exofacial ferricyanide reduction reaction?

-P9L158: "pH drift" connotes instrument drift to me, not pH changes due to cellular activity

-P9L169: change "on differences"

-P9L174: Need more information on the Bonferroni correction.

-P11L200: is the Bonferroni correction the same as the "Post hoc LSD comparison" mentioned here? I don't think it is? What is this comparison?

-P11L213: change "access" to "assess"

-P11L216: what does "interplayed" signify?

-P12L230: is the peak the same as the plateau mentioned earlier?

-P12L231: what do you mean by "assayed"?

-P13L247: how was the pH compensation point identified?

-P13L260-261: not sure what "comparative photosynthetic rates" means

-P15L293: does "inorganic carbon" here mean both carbonate and bicarbonate?

-P15L303: change to "increased the redox"

-P15L304: misspelled extracellular

-P16L315: change to "as the $CO_2$"

-P16L317: change to "with a strong"

-P16L319: change to "the red macroalgae"

-P17L340: change to "the potential mechansims"

-P17L342: change to "are hampered"

-P17L348: change to "growth"

---

## Referee Comment (RC2) · Anonymous Referee #2 · 14 Mar 2018

This manuscript reports results of experiments which aim to investigate the link between P availability and the C uptake by S. costatum diatoms. While apparently interesting interactions were observed, insufficient detail is provided about the methods, and I have reservations about the suitability of the statistical analysis employed.

Major Comments

The introduction would benefit from adding hypotheses.

The methods section has a rather low amount of detail for each of the methods presented, with details of instrument manufacturers and models, and references frequently missing. In particular, there is no mention of how cells were counted, and normalising

this is an important aspect of many of the measurements.

I am also not convinced that 3 replicates of each treatment is sufficient, at least not for the parametric statistical testing that is employed.

The results section does not report what the actual values of the measured parameters were, only the results of statistical tests for differences between treatments.

There is rather limited discussion of the mechanisms behind each of the effects observed.

Minor Comments

Not all of the figures are referred to in the text, or at least not in the correct order (there is no Fig. 3 reference between the first reference for Fig. 2 and that for Fig. 4).

Line 10: Define rETR the first time it is used.

Line 43: This should say 'limiting', not 'limited'.

Line 48: Give the name in full the first time it is used, and where it is used at the start of a sentence.

Lines 54-60: Please define all these acronyms the first time they are used.

Line 88: I don't think the units given here for irradiance are correct (micromoles per m squared).

---

## Author Comment (AC1) · 21 Mar 2018

Reviewer 1 The authors present a manuscript detailing culture studies of a common algae species under varying CO2 and phosphorus concentrations. Algal blooms can draw down dissolved CO2 to very low levels, and some species have developed mechanisms to compensate for decreased CO2 availability. Some of these mechanisms may be influenced by the presence or absence of bio-available phosphorus, leading to the study design of varying phosphorus levels across test populations of two CO2 levels. While there is probably a compelling study underlying this manuscript, in my opinion there are too many flaws as presented to encourage publication. I feel compelled to point out here that my background is in seawater carbonate chemistry, and my knowledge of cellular biochemical processes is limited. However, based solely on the description of inorganic carbon system measurements I would advise rejecting this manuscript until serious revisions have been done. I will present some major comments below, followed by more minor concerns.

Response: We appreciate these comments and believe the manuscript has been largely improved by responding to all comments raised by the reviewer.

MAJOR COMMENTS -The Materials and Methods section, in particular the sections pertaining to pH and alkalinity measurements, is totally inadequate. Is the pH system an electrode-type system? What pH scale are measurements presented in? How was the pH system calibrated? How were alkalinity titrations performed? No information is presented. These questions are especially critical in the calculation of DIC from pH and alkalinity (P7L129-131), which is very sensitive to relatively minor changes in pH and alkalinity. With no information about the quality of pH and alkalinity measurements, the results of this analysis are impossible to interpret.

Response: We appreciate these comments and apologize for missing these details. The text has been clarified to "The pHNBS was measured by a pH meter (pH 700, Eutech Instruments, Singapore) that was equipped with an Orion$^{®}$ 8102BN Ross combination electrode (Thermo Electron Co., USA) and calibrated with standard National Bureau of Standards (NBS) buffers (pH = 4.01, 7.00, and 10.01 at 25.0 oC; Thermo Fisher Scientific Inc., USA). Total alkalinity (TAlk) was determined at 25.0 oC by Gran acidimetric titration on a 25-ml sample with a TAlk analyzer (AS-ALK1, Apollo SciTech, USA), using the precision pH meter and an Orion$^{®}$ 8102BN Ross electrode for detection. To ensure the accuracy of TAlk, the TAlk analyser was regularly calibrated with certified reference materials from Andrew G. Dickson's laboratory (Scripps Institute of Oceanography, U.S.A.) at a precision of $\pm$ 2 $\mu$mol kg-1." at P8L127-135.

-Besides using the barely-described pH system, how did the authors know the CO2

levels of their cultures? Also, adding phosphate to the cultures, at concentrations rang-
ing from 0.5-10 _mol/kg adds a potent buffering agent, as monosodium phosphate has
a pKa around 7. How did the authors alter or maintain the pH in the cultures? Were
the cultures open to the ambient atmosphere?

Response: We apologize for missing these details. The text has been clarified to "The
two levels of pH (8.20 and 8.70) were obtained by aerating the ambient air and pure
nitrogen (99.999%) till the target value, and were then maintained with a buffer of 50
mM tris (hydroxymethyl) aminomethane-HCl. The cultures were open to the ambient
atmosphere and the variation of culture pH was below $\pm$ 0.02 unites during the two
hours of pH treatment. $CO_2$ level in seawater was calculated via CO2SYS (Pierrot
et al., 2006) based on measured pH and TAlk, using the equilibrium constants of K1
and K2 for carbonic acid dissociation (Roy et al., 1993) and the KSO4- dissociation
constant from Dickson (1990)." at P8L119-126.

-As previously mentioned, my knowledge of some of the biochemical processes pre-
sented here is minimal, and the Introduction did little to help readers like myself. There
seem to be connections between plasma membrane redox activity, CAext, rETR, but
the manuscript does not explain them. Some terms (i.e. rETR) are presented with
no explanation or definition. Thus the reason for some of the analyses presented was
unclear to me. What did measuring the chlorophyll fluorescence inform? The cultures
were initiated at the same cell density, but surely the cell density varied between cul-
tures after the treatment period- how was this accounted for?

Response: We apologize for the confusion. The connections between plasma mem-
brane redox activity, CAext, rETR were explained in Discussion. To make readers
know them earlier, we have generally explained the connections in Methods. It reads
"Chlorophyll fluorescence was measured with a pulse modulation fluorometer (PAM-
2100, Walz, Germany) to assess electron transport in photosystem II and the possible
connection between electron transport and redox activity of the plasma membrane." at
P9L141-143. rETR has been defined as "relative electron transport" in abstract where

it first appeared at P2L11. In terms of cell density, it did not vary during the two hours of pH treatment as diatom cells usually proliferate at night. This information has been added to the text and it reads "The cell density did not vary during the two hours of pH treatment." at P8L116-117.

-In the Results section the authors present their statistical findings in the form $F(1,20)=XX$ or $F(4,20)=XX$. I'm assuming these are the results of the two-way ANOVA test mentioned in the 2.8 Statistical Analysis section, but no explanation is given as to what is signified. Are the numbers in parentheses indicating degrees of freedom? What is the threshold for significance?

Response: Yes, the numbers in parentheses indicating degrees of freedom. The threshold for significance is $P < 0.05$, which was stated in section 2.8.

-The Results section is extremely repetitive. Much of the information presented could be more effectively summarized in a table.

Response: The statistical outcomes have been presented in tables as suggested.

MINOR COMMENTS -The English usage in much of the manuscript could be improved. I will note some points below.

Response: We appreciate the constructive comments very much and revised the manuscript based on all the comments.

-Define rETR in the Abstract (P2L10)

Response: rETR has been defined as "relative electron transport".

-P2L3 and throughout: define the abbreviations when first used: $CO_2$, DIC, $HCO_3$-, rETR

Response: They have been corrected to "carbon dioxide ($CO_2$), relative electron transport rate (rETR), bicarbonate ($HCO_3$-), dissolved inorganic carbon (DIC)".

-P2L16 change "is" to "was"

Response: Corrected.

-P3L26 change to "the marine biological"

Response: Corrected.

-P3L31 need a different word than compelling

Response: It has been changed to "key".

-P3L33 and throughout: don't finish sentences with "etc".

Response: It has been changed to " and so forth".

-P3L34 change "could" to "can".

Response: Corrected.

-P3L36 misselled "dominate"

Response: Corrected

-P3L37 P3L40-41 How is RUBISCO important? What is it, an enzyme?

Response: It has been revised to "Diatoms' ribulose-1,5-bisphosphate carboxylase/oxygenase (RUBISCO), catalyzing the primary chemical reaction by which CO2 is transformed into organic carbon, has a relatively low affinity for CO2 and is commonly less than half saturated under current CO2 levels in seawater (Hopkinson & Morel, 2011)." at P4L45-49.

-P4L45-48: the carbon concentrating mechanisms are named but not explained. A reader like myself has no way to know what "multiple carbon anhydrase, assumed C4-type pathway" represents

Response: It has been revised to "diatoms have evolved various inorganic carbon acquisition pathways and CO2 concentrating mechanisms (CCMs), for instance, active

transport of HCO3-, the passive influx of CO2, multiple carbonic anhydrase (including both common ($\alpha$, $\beta$, $\gamma$) and unusual ($\delta$, $\zeta$) families that carries out the fast interconversion of CO2 and HCO3-), assumed C4-type pathway (using phosphoenolpyruvate to capture more CO2 in the periplastidal compartment), to increase the concentration at the location of Rubisco and thus the carbon fixation(Hopkinson & Morel, 2011; Hopkinson et al., 2016)." at P5L51-57.

-P4L54 define CCMs (CO2 concentrating mechanisms, right?)

Response: It was defined in the abstract. We have defined it again at line P5L52.

-P4L57: keep consistent units between discussions of CO2 or pCO2. Discuss either CO2 concentration or partial pressure. Reader has no way to compare 5 mol/L CO2 to a pCO2 of 1800 _atm.

Response: It has been revised to "extracellular carbonic anhydrase activity in S. costatum was only induced when CO2 concentration was less than 5 $\mu$mol L-1 while Rost et al. (2003) reported that activity of extracellular CA could be detected even when CO2 concentration was 27 $\mu$mol L-1." at P5L67-69.

-P5L69: cite the refernces yourself, don't refer to references therein

Response: It has been revised to "phosphorus acquisition, utilization and storage (Lin et al., 2016; Gao et al., 2018)." at P6L80.

-P5L70: what is the relationship?

Response: It has been revised to "Some studies show the essential role of phosphorus in regulating inorganic carbon acquisition in green algae." at P6L81-82.

-P5L75-76: remove "the capacity of"

Response: Corrected.

-P5L78-79: all these mechanisms/pathways! How do they interrelate?

Response: It has been revised to "we aimed to test this hypothesis by investigating the variation of CCMs (including active transport of $HCO_3^-$ and carbonic anhydrase activity) and photosynthetic rate under five levels of phosphate and two levels of $CO_2$ conditions. We also measured redox activity of plasma membrane as it is deemed to be critical to activate carbonic anhydrase (Nimer et al., 1998)." at P6L89-95.

-P5L82: as $CO_2$ is removed by diatom growth, the inorganic carbon equilibria will shift to convert $HCO_3^{2-}$ to $CO_2$. How do the kinetics of this potentially affect this study?

Response: It is exactly true that the shift from $HCO_3^-$ to $CO_2$ will occur as $CO_2$ is removed by diatom's growth and it usually leads to the increase of $OH^-$ and pH in seawater because the dissolution rate of $CO_2$ from the atmosphere cannot catch up with its removal rate (Gao et al., 1993; Hansen, 2002). However, the pH in the cultures was relatively stable due to the addition of tris (hydroxymethyl) aminomethane-HCl buffer, which resulted in stable $CO_2$ levels in the cultures. And this is what we aimed to achieve so that the culures could be conducted under two $CO_2$ levels: one is the aimbient level (12.6 $\mu$mol L-1) and the other is $CO_2$ limiting level (2.8 $\mu$mol L-1).

The text has been clarifed to "The cultures were open to the ambient atmosphere and the rise of culture pH was below 0.02 unites (corresponding to the rise of $CO_2$ less than 0.7 and 0.2 $\mu$mol L-1 for pH 8.20 and 8.70 treatments, respectively) during the two hours of pH treatment." at P8L120-123.

Gao K, Aruga Y, Asada K, et al. Enhanced growth of the red alga Porphyra yezoensis Ueda in high $CO_2$ concentrations. Journal of Applied Phycology, 1991, 3(4): 355-362.

Hansen PJ. 2002. Effect of high pH on the growth and survival of marine phytoplankton: implications for species succession. Aquatic Microbial Ecology 28: 279-288.

-P6L105: what does "algae after in light" mean?

Response: It has been clarified to "where FM' is the maximal fluorescence levels from algae in the actinic light after application a saturating pulse" at P9L146-147.

-P6L108: change to "photosynthetic and respiration rates"

Response: Corrected.

-P7L114-116: why measure photosynthesis for 5 minutes but respiration for 10?

Response: The oxygen variation rate due to dark respiration is slower than that caused by photosynthesis, so more times are needed for dark respiration measurement. It has been explained "To measure dark respiration rate, the samples were placed in darkness and the decrease of oxygen content within ten minutes was defined as dark respiration rate given the slower oxygen variation rate for dark respiration." at P9L156-159.

-P7L126: what is "Ci-saturated maximum rate"?

Response: It has been revised to "DIC-saturated maximum rate" at P10L169.

-P8L149: by "samples" do you mean the diatom cells?

Response: Yes, it has been changed to "cells" at P11L194.

-P9L156: what is the exofacial ferricyanide reduction reaction?

Response: Exofacial means extracellular. The text has been clarified to "The redox activity of plasma membrane was assayed by monitoring the change in ferricyanide $K_3Fe(CN)_6$ concentration that accompanied reduction of the ferricyanide to ferrocyanide. The ferricyanide $[K_3Fe(CN)_6]$ cannot penetrate intact cells and has been used as an external electron acceptor (Nimer et al., 1998; Wu & Gao, 2009). Stock solutions of $K_3Fe(CN)_6$ were freshly prepared before use. Five mL of samples were taken after two hours of incubation with 500 $\mu$mol $K_3Fe(CN)_6$ and centrifuged at 4000 g for 10 min (20oC). The concentration of $K_3Fe(CN)_6$ in the supernatant was measured spectrophotometrically at 420 nm (Shimadzu UV-1800, Kyoto, Japan). The decrease of $K_3Fe(CN)_6$ during the two hours of incubation was used to assess the rate of extracellular ferricyanide reduction (Nimer et al., 1998)." at P11L196-206.

-P9L158: "pH drift" connotes instrument drift to me, not pH changes due to cellular Activity

Response: To clarify it, the text has been revised to "Cell-driving pH drift experiment" at P12L207.

-P9L169: change "on differences"

Response: It has been corrected to "to assess the effects of CO2 and phosphate on net photosynthetic rate" at P12L217-218.

-P9L174: Need more information on the Bonferroni correction. -P11L200: is the Bonferroni correction the same as the "Post hoc LSD comparison" mentioned here? I don't think it is? What is this comparison?

Response: The Bonferroni test uses a straight-forward t test but then evaluates that t at $\alpha$ = 0.05/c, where c is the number of comparisons. Some of the post-hoc comparisons may not be appropriate for repeated-measure ANOVA while Bonferroni is the best reliable one (Ennos, 2007). The text has been revised to "Bonferroni was conducted for post hoc investigation as it is the best reliable post hoc test for repeated measures ANOVA (Ennos, 2007)"at P12L223-224.

Ennos A R. Statistical and Data Handling Skills in Biology. Pearson Education, 2007, p.96.

-P11L213: change "access" to "assess"

Response: Corrected.

-P11L216: what does "interplayed" signify?

Response: It has been changed to "interacted".

-P12L230: is the peak the same as the plateau mentioned earlier?

Response: Yes.

-P12L231: what do you mean by "assayed"?

Response: It has been changed to "assessed".

-P13L247: how was the pH compensation point identified?

Response: As mentioned in section 2.7, "the pH compensation point was obtained when there was no a further increase in pH." at P12L211-212.

-P13L260-261: not sure what "comparative photosynthetic rates" means

Response: It has been corrected to "showed similar photosynthetic rates for the lower and higher CO2 treatments." at P17L313-314.

-P15L293: does "inorganic carbon" here mean both carbonate and bicarbonate?

Response: No, it means both $CO_2$ and $HCO_3^-$ because $CO_3^{2-}$ cannot be used for photosynthesis. The text has been revised to " inorganic carbon ($CO_2$ and $HCO_3^-$)" P18L338.

-P15L303: change to "increased the redox"

Response: Corrected.

-P15L304: misspelled extracellular

Response: Corrected.

-P16L315: change to "as the CO2"

Response: Corrected.

-P16L317: change to "with a strong"

Response: Corrected.

-P16L319: change to "the red macroalgae"

Response: Corrected.

-P17L340: change to "the potential mechansims"

Response: Corrected.

-P17L342: change to "are hampered"

Response: Corrected.

-P17L348: change to "growth"

Response: Corrected.

---

## Author Comment (AC2) · 21 Mar 2018

Anonymous Referee #2 This manuscript reports results of experiments which aim to investigate the link between P availability and the C uptake by S. costatum diatoms. While apparently interesting interactions were observed, insufficient detail is provided about the methods, and I have reservations about the suitability of the statistical analysis employed.

Response: We appreciated these comments and have improved the manuscript by responding to the comments.

[Figure]

Major Comments The introduction would benefit from adding hypotheses.

Response: We did state our hypothesis at line and it reads "Based on the connection between phosphorus and carbon metabolism in diatoms (Brembu et al., 2017), we hypothesize that phosphorus enrichment could enhance inorganic carbon utilization and hence maintain high rates of photosynthesis and growth in S. costatum under $CO_2$ limitation conditions." at P6L85-89.

The methods section has a rather low amount of detail for each of the methods presented, with details of instrument manufacturers and models, and references frequently missing. In particular, there is no mention of how cells were counted, and normalizing this is an important aspect of many of the measurements.

Response: We appreciate these comments and have added more details to the Methods. In terms of cell counting, it has been clarified to "Cell density was determined by direct counting with an improved Neubauer haemocytometer (XB-K-25, Qiu Jing, Shanghai, China)." at P7L110-111.

I am also not convinced that 3 replicates of each treatment is sufficient, at least not for the parametric statistical testing that is employed.

Response: We agree that a higher replication would strengthen the study. However, we had to reduce the replication to obtain reliable data as we had 10 treatments for each measurement, indicating 30 samples (10×3) for each measurement. Three-replicate is fine for parametric statistical tests and it were used in many studies (Riebesell et al., 2007; Gao et al., 2012; Walworth et al., 2016; Hong et al., 2017).

Hong H, Shen R, Zhang F, et al. The complex effects of ocean acidification on the prominent N2-fixing cyanobacterium Trichodesmium. Science, 2017, 356(6337): 527-531.

Riebesell U, Schulz K G, Bellerby R G J, et al. Enhanced biological carbon consumption in a high CO2 ocean. Nature, 2007, 450(7169): 545-548.

Gao K, Xu J, Gao G, et al. Rising CO2 and increased light exposure synergistically reduce marine primary productivity. Nature Climate Change, 2012, 2(7): 519-523.

Walworth N G, Fu F X, Webb E A, et al. Mechanisms of increased Trichodesmium fitness under iron and phosphorus co-limitation in the present and future ocean. Nature Communications, 2016, 7: 12081.

The results section does not report what the actual values of the measured parameters were, only the results of statistical tests for differences between treatments.

Response: In our previous manuscripts, we were informed by some reviewers that the report of actual values was unnecessary as readers could see them from tables and figures. We anyhow added some actual values to the Results section at P13L226-P16L302.

There is rather limited discussion of the mechanisms behind each of the effects observed.

Response: We agree that we did not discuss the molecular mechanisms for the effects observed. We did not do it as our study did not refer to molecular measurements. Instead, we compared our results to those of similar studies, explained the meaning of our finding, tested our hypothesis by integrating the measured parameters, and finally produced take-home massage "P enrichment could induce activity of extracellular carbonic anhydrase and direct utilization of HCO3- in S. costatum to help overcome the CO2 limitation, as well as increasing photosynthetic pigment content and rETR to provide required energy." in the Conclusion section. Honestly, we have no idea of how to improve the discussion of the mechanisms behind each of the effects observed and hope to hear more specific suggestions.

Minor Comments Not all of the figures are referred to in the text, or at least not in the correct order (there is no Fig. 3 reference between the first reference for Fig. 2 and that for Fig. 4).

Response: We think all the figures were referred to in the text in order. We did refer to Fig. 3 between Fig. 2 and Fig. 4 at P14L251.

Line 10: Define rETR the first time it is used.

Response: It has been defines as "relative electron transport rate".

Line 43: This should say 'limiting', not 'limited'.

Response: Corrected.

Line 48: Give the name in full the first time it is used, and where it is used at the start of a sentence.

Response: Corrected throughout the text.

Lines 54-60: Please define all these acronyms the first time they are used.

Response: Corrected.

Line 88: I don't think the units given here for irradiance are correct (micromoles per m squared).

Response: It has been corrected to "$\mu$mol photons m-2 s-1" at P7L103.

---

## Referee Report (RR1)

Review of Gao et al., "Regulation of inorganic carbon acquisition in a red tide alga (Skeletonema costatum): the importance of phosphate availability.

This is a review of a resubmitted, revised manuscript, which I originally reviewed.  The authors should be commended for taking the original reviews from myself and another reviewer and making substantial changes and additions to their manuscript.  The revised manuscript is improved, and I believe is suitable for publications following some minor revisions I list below.

-One overarching question still remains to me: if P concentrations are important to these carbon concentrating mechanisms, which in turn promote bloom formation, what is the likelihood that these higher P concentrations will be available during a bloom, as P is drawn down along with $CO_2$ during a bloom?  It seems that the P would become as unavailable as $CO_2$ during bloom formation, and thus not able to be utilized by in these CCMs, unless nutrient conditions are quite eutrophic.  Perhaps the authors can speculate on this question in the discussion?

Minor Comments:

-P5L51: these common and unusual family notations (alpha, beta, sigma etc.) are not really explained, and unfamiliar to me

P5L54: remove "." after "fixation"

-P5L67 remove "in" from "that in S."

-P6L71 define ATP

-P6L90- change to "Our study provides helpful…"

-P7L98- change to "set as 200…"

-P7L109- change to "blooms"

-P8L114 change "till" to "until"

-P8L116 changes to "units"

-P8L122-134- these lines are all indented and do not need to be

-P8L130-132- this is a direct copy of the previous statement in P8L118-120

-P9L135- what is "photosystem II"?

-P9L139 change to $e^{-1}$

-P12L203- change to "no further"

-P18L337- change to "exactly the same"

-P18L348- change to "based on experiments"

-P18L352- change to "directly, and whose photosynthesis…"

-P19L366- change to "In the development of red tides, the pH in seawater can be…"

-P19L376- change to "S. costatum in order to overcome …"

-P20L382- change to "overcome $CO_2$ limitation…"

-P20L385- change to "mechanisms that help *S. costatum* dominate algal blooms."

---

## Author Response (AR2)

Review of Gao et al., "Regulation of inorganic carbon acquisition in a red tide alga (Skeletonema costatum): the importance of phosphate availability.

This is a review of a resubmitted, revised manuscript, which I originally reviewed. The authors should be commended for taking the original reviews from myself and another reviewer and making substantial changes and additions to their manuscript. The revised manuscript is improved, and I believe is suitable for publications following some minor revisions I list below.

Response: We appreciate these comments very much.

-One overarching question still remains to me: if P concentrations are important to these carbon concentrating mechanisms, which in turn promote bloom formation, what is the likelihood that these higher P concentrations will be available during a bloom, as P is drawn down along with CO2 during a bloom? It seems that the P would become as unavailable as CO2 during bloom formation, and thus not able to be utilized by in these CCMs, unless nutrient conditions are quite eutrophic. Perhaps the authors can speculate on this question in the discussion?

Response: We appreciate this constructive suggestion. Although P is replete in eutrophic waters at the early stage of algal blooms, P limitation may occur at the late stage of algal bloom, which leads to shift of dominating algae. We have added this point to the text "The CCMs of *S. costatum* are hampered under P limiting conditions and only function when P is replete. This finding may explain why diatoms could overcome carbon limitation and dominate red tides when P is replete and as well as the shift from diatoms to dinoflagellates when P is limiting (Mackey et al., 2012)" at P20L388-392.

Minor Comments:

-P5L51: these common and unusual family notations (alpha, beta, sigma etc.) are not really explained, and unfamiliar to me

Response: It has been calrified to "multiple carbonic anhydrase (including both common ($\alpha$, $\beta$, $\gamma$, found in all algae) and unusual ($\delta$, $\zeta$, found only in diatoms) families that carries out the fast interconversion of $CO_2$ and $HCO_3^-$)" at P5L51-15.

P5L54: remove "." after "fixation"

Response: Corrected.

-P5L67 remove "in" from "that in S."

Response: Corrected.

-P6L71 define ATP

Response: It has been revised to "adenosine-triphosphate (ATP)".

-P6L90- change to "Our study provides helpful…"

Response: Corrected.

-P7L98- change to "set as 200…"

Response: Corrected.

-P7L109- change to "blooms"

Response: Corrected.

-P8L114 change "till" to "until"

Response: Corrected.

-P8L116 changes to "units"

Response: Corrected.

-P8L122-134- these lines are all indented and do not need to be

Response: Corrected.

-P8L130-132- this is a direct copy of the previous statement in P8L118-120

Response: We apologize for the repetition. These lines have been removed.

-P9L135- what is "photosystem II"?

Response: It has been clarified to "photosystem II (the first protein complex in the light-dependent reactions of photosynthesis)".

-P9L139 change to e-1

Response: We think it would be better to keep it as here e$^-$ means electron with negative charge and this is the usual expression for electron transport rate in photosystems (Alderkamp et al, 2012; Perkins et al., 2018).

Alderkamp, A. C., Kulk, G., Buma, A. G., Visser, R. J., Van Dijken, G. L., Mills, M. M., & Arrigo, K. R. (2012). The effect of iron limitation on the photophysiology of *Phaeocystis antarctica* (Prymnesiophyceae) and *Fragilariopsis cylindrus* (Bacillariophyceae) under dynamic irradiance. Journal of Phycology, 48(1), 45-59.

Perkins, R., Williamson, C., Lavaud, J., Mouget, J. L., & Campbell, D. A. (2018). Time-dependent upregulation of electron transport with concomitant induction of regulated excitation dissipation in *Haslea* diatoms. Photosynthesis Research, 1-12.

-P12L203- change to "no further"

Response: Corrected.

-P18L337- change to "exactly the same"

Response: Corrected.

-P18L348- change to "based on experiments"

Response: Corrected.

-P18L352- change to "directly, and whose photosynthesis…"

Response: Corrected.

-P19L366- change to "In the development of red tides, the pH in seawater can be…"

Response: Corrected.

-P19L376- change to "S. costatum in order to overcome …"

Response: Based on the previous comment, it has been revised to "This finding may explain why diatoms could overcome carbon limitation and dominate red tides when P is replete and as well as the shift from diatoms to dinoflagellates when P is

limiting (Mackey et al., 2012)." at P20L389-392.

-P20L382- change to "overcome CO2 limitation…"

  Response: Corrected.

-P20L385- change to "mechanisms that help *S. costatum* dominate algal blooms."

  Response: Corrected.

The authors grew the model diatom Skeletonema costatum in seawater media at pH 8.2 at 5 [PO4-].They shifted samples to pH8.2 (no change) or to pH 8.7 and measured the photosynthetic responses to a matrix of imposed [PO4-] and [CO2]. They do not state (although they have the data) the [DIC] under their treatments, and I question their protocol for lowering [CO2] which depends on blocking equilibration between the culture suspension and the headspace, if any. I offer suggestions for major (although easy) redesign of the figures that might help convey the findings.
best regards, Doug Campbell

  Response: We appreciate the reviewer's comments and the manuscript has been revised by responding to these comments.

Figure 1:

This figure should be an X-Y plot of photosynthesis (A) or respiration (B) vs. umol Phosphate L$^{-1}$. Each graph should have 2 series, a low carbon dioxide and a high carbon dioxide.

  Do not erect unnecessary levels of coding; 'AC'? 'LC'?? Why? Just give the CO2 concentrations.

  Response: Corrected.

Fig. 2 should be panel C of Figure 1, same comments as above.

  Response: Corrected.

rETR umol e- m-2 s-1 is not a meaningful unit for a suspension of cells. Think about what is being measured and how.

Same comment regarding XY plot as Figure 1

  Response: We thank the reviewer for this constructive comment. By following the method of Perkins et al (2018), the absolute photosynthetic electron transport rate through PSII (ETR, μmol e$^{-}$ PSII$^{-1}$s$^{-1}$) can be estimated by the following equation: ETR = E $\times \sigma_{PSII}$' $\times /\Phi_{PSII}/((F_M-F_0)/ F_M)$. However, we do not have a fluorometer that can measure $\sigma_{PSII}$'. To estimate the absolute ETR, we would like to use the following equation: ETR (ETR, μmol e$^{-}$ (mg Chl *a*) $^{-1}$s$^{-1}$) = 0.5 $\times$E $\times\Phi_{PSII}$ $\times\bar{a}^{*}$ (Dimier et al., 2009; Alderkamp et al., 2012), where $\Phi_{PSII}$ (dimensionless) is the PSII photochemical efficiency, E (μmol photons m$^{-2}$ s$^{-1}$) is the ambient light density and $\bar{a}^{*}$ is Chl *a*–specific absorption coefficient (m$^{-2}$ (mg Chl *a*)$^{-1}$). Since $\bar{a}^{*}$ is light-dependent, we used the value of 0.0138 based on Lefebvre et al's (2007) study in which the light density is very close to ours.

Alderkamp, A. C., Kulk, G., Buma, A. G., Visser, R. J., Van Dijken, G. L., Mills, M. M., & Arrigo, K. R. (2012). The effect of iron limitation on the photophysiology of *Phaeocystis antarctica* (Prymnesiophyceae) and *Fragilariopsis cylindrus* (Bacillariophyceae) under dynamic irradiance. Journal of Phycology, 48(1), 45-59.

Dimier, C., Brunet, C., Geider, R., & Raven, J. (2009). Growth and photoregulation dynamics of the picoeukaryote *Pelagomonas calceolata* in fluctuating light. Limnology and Oceanography, 54(3), 823-836.

Lefebvre, S., Mouget, J. L., Loret, P., Rosa, P., & Tremblin, G. (2007). Comparison between fluorimetry and oximetry techniques to measure photosynthesis in the diatom *Skeletonema costatum* cultivated under simulated seasonal conditions. Journal of Photochemistry and Photobiology B: Biology, 86(2), 131-139.

Perkins, R., Williamson, C., Lavaud, J., Mouget, J. L., & Campbell, D. A. (2018). Time-dependent upregulation of electron transport with concomitant induction of regulated excitation dissipation in *Haslea* diatoms. Photosynthesis Research, 1-12.

Fig. 4 same comment

Response: Corrected.

Fig. 5, OK, now we switch the rules and plot photosynthesis vs. DIC, with series for [PO4-].

Response: Yes. This is what we did.

Fig. 6, same comment as Fig. 1. Do not put a quantitative data series as a categorical axes. 0.05 to 10 umol PO4- l-1 are not different categories, they are arbitrary measurement points along a (potentially) continuous axes of PO4-.

Use a log scale if necessary.

Response: Corrected.

Fig. 7, same comment.

Response: Corrected.

Materials & Methods:

Lines 101-103

This is an odd, incomplete way of expressing the applied DIC treatment. What was the [DIC] under the two treatments? A shift from bubbling with ambient air (outdoor? indoor?) pH 8.2 to ph 8.2 (no change) or pH 8.7 (increase) would lead to increased dissolution of head space $CO_2$ into water at pH 8.7, with a concomitant increase in [DIC] if the cell suspension is allowed to equilibrate with a headspace. If not headspace is provided the suspension will be in [$CO_2$] deficit as pH equilibration drives [DIC] to [$HCO_3^-$], lowering [$CO_2$]. Then, any exposure to gas with ambient $CO_2$ will lead to uptake.

Response: The concentrations of DIC were 2109 $\pm$ 36 and 1802 $\pm$ 38 μmol (kg seawater)$^{-1}$, respectively. As mentioned in the text, the cultures were open to the ambient atmosphere but this did not lead to the increase of DIC or $CO_2$. Instead, there

was a slight decrease in $CO_2$ because of the algal photosynthesis. The text has been clarified to "Afterwards, cells were resuspended in fresh media with two levels of pH (8.20 and 8.70, respectively corresponding to ambient $CO_2$ (12.6 µmol $L^{-1}$, AC) and low $CO_2$ (2.8 µmol $L^{-1}$, LC) under corresponding phosphate levels for two hours before the following measurements, with a cell density of $1.0 \times 10^6$ $mL^{-1}$. The concentrations of DIC were 2109 $\pm$ 36 and 1802 $\pm$ 38 µmol (kg seawater)$^{-1}$, respectively." at P37L102-107 and "The cultures were open to the ambient atmosphere and the rise of culture pH due to algal photosynthesis was below 0.02 units (corresponding to the decrease of $CO_2$ less than 0.7 and 0.2 µmol $L^{-1}$ for pH 8.20 and 8.70 treatments respectively) during the two hours of pH treatment." atP8L118-121.

Line 140: the rETR units are algebraically correct, but are not meaningful in the sense of a cell suspension.

Response: This has been corrected and please see the response above.